# Continuous Bayesian Model Selection for Multivariate Causal Discovery

Anish Dhir [* 1]   Ruby Sedgwick [* 1 2]   Avinash Kori [1]   Ben Glocker [1]   Mark van der Wilk [3]

## Abstract

Current causal discovery approaches require restrictive model assumptions in the absence of interventional data to ensure structure identifiability. These assumptions often do not hold in real-world applications leading to a loss of guarantees and poor performance in practice. Recent work has shown that, in the bivariate case, Bayesian model selection can greatly improve performance by exchanging restrictive modelling for more flexible assumptions, at the cost of a small probability of making an error. Our work shows that this approach is useful in the important multivariate case as well. We propose a scalable algorithm leveraging a continuous relaxation of the discrete model selection problem. Specifically, we employ the Causal Gaussian Process Conditional Density Estimator (CGP-CDE) as a Bayesian nonparametric model, using its hyperparameters to construct an adjacency matrix. This matrix is then optimised using the marginal likelihood and an acyclicity regulariser, giving the maximum a posteriori causal graph. We demonstrate the competitiveness of our approach, showing it is advantageous to perform multivariate causal discovery without infeasible assumptions using Bayesian model selection.

## 1. Introduction

In many systems, such as protein signalling networks, variables causally relate to each other, since changing a variable only modifies certain variables (Sachs et al., 2005). Uncovering these unique underlying causal structures from data can allow us to gain new insights in a wide range of fields, from

Biology (Sachs et al., 2005) to Medicine (Feuerriegel et al., 2024) to Economics (Hicks et al., 1980). There are two main approaches for learning the ground truth causal structure from data. The first approach assumes a restricted model class for the data generating process (Peters et al., 2017, Ch. 4), but its guarantees often fail when these assumptions do not hold. The second approach requires interventional data for all variables, which can be costly, ethically challenging, or even impossible (Li et al., 2019). Consequently, learning a unique causal structure often relies on impractical assumptions, limiting its real-world applicability.

Recent work has shown that, in the absence of interventional data and in the bivariate case, Bayesian model selection can more accurately infer the unique causal structure of a system (Dhir et al., 2024). Unlike previous methods, including other Bayesian methods[1], which impose constraints on the function class, this framework allows the use of more flexible model classes. This comes at a cost of loosened identifiability guarantees but with the ability to posit more realistic assumptions. However, Dhir et al. (2024) only study the bivariate case, while most applications require understanding causal relationships between multiple variables. It is thus an open question whether the advantages of this method hold in the multivariate case. Furthermore, the approach in Dhir et al. (2024) requires computing and comparing the posteriors of all possible causal graphs. As the number of graphs grows super-exponentially with the number of variables, this becomes computationally intractable for scalability.

A potential solution lies in continuous optimisation causal discovery approaches (Zheng et al., 2018; Nazaret et al., 2024). These approaches interpret causal discovery as a single continuous optimisation problem that is amenable to gradient based methods. The emphasis of this line of work has been either on scaling to a larger number of variables or on loosening model assumptions. Our contribution tackles the latter. We argue that the potential advantages of Bayesian methods enable them to outperform their non-Bayesian counterparts.

We investigate whether and how well Bayesian model se-

*Equal contribution [1]Department of Computing, Imperial College London, London, UK [2]Xyme, Oxford, UK [3]Department of Computer Science, University of Oxford, Oxford, UK. Correspondence to: Anish Dhir <anish.dhir13@imperial.ac.uk>.

*Proceedings of the 42nd International Conference on Machine Learning*, Vancouver, Canada. PMLR 267, 2025. Copyright 2025 by the author(s).

---

[1]These methods are usually concerned with inferring a distribution instead of a single causal structure.

lection can recover the ground truth causal structure in multivariable systems. We use a model based on the Gaussian Process conditional density estimator (GP-CDE) (Titsias & Lawrence, 2010; Lalchand et al., 2022) that is more flexible than traditional identifiable models that rely on restrictive model assumptions. Interpreting the model hyperparameters as a graph, we optimise over them using a Bayesian model selection-based loss and a regulariser to enforce acyclicity, ultimately producing the *maximum a posteriori* (MAP) causal graph. Our experiments address two critical questions: (1) How much performance is lost compared to enumerating every causal graph? (2) How does our scalable Bayesian model selection compare to existing multivariate causal discovery methods in terms of performance? Our results show that Bayesian model selection outperforms methods enforcing strict identifiability even at larger scales. While it introduces a small error probability, the flexibility afforded enables more realistic and practical multivariable causal discovery, advancing toward real-world applications.

## 2. Background

Here, we outline the assumptions relevant to the task of causal discovery. Additional background is in Appendix A.

### 2.1. Causal Model

We assume that data $\mathbf{X} \in \mathbb{R}^{N \times D}$ can be explained by a *Causal Model*, which we refer to as $\mathcal{M}_\mathcal{G}$. We make the common assumption that there are no hidden confounders.

**Definition 2.1.** (Consistent with Pearl (2009) but using notation from Dhir et al. (2024)) A *Causal Model* $\mathcal{M}_\mathcal{G} := (\mathcal{G}, \mathcal{C}_\mathcal{G})$ is defined as a tuple containing a DAG $\mathcal{G}$ with vertex set $\mathcal{V}$ and edge set $\mathcal{E}$, along with a set of conditional distributions for each variable $\mathcal{C}_{i|\mathrm{PA}_\mathcal{G}(i)}$, where $\mathrm{PA}_\mathcal{G}(i)$ denotes the parent index set of node index $i$ in $\mathcal{G}$. The set of all possible conditionals is the Cartesian product $\mathcal{C}_\mathcal{G} := \prod_{i \in \mathcal{V}} \mathcal{C}_{i|\mathrm{PA}_\mathcal{G}(i)}$. The set of all possible conditional distributions induces a set of joint distributions denoted $\mathcal{F}_\mathcal{G}$.

An element of $\mathcal{C}_\mathcal{G}$ induces a joint distribution over $D$ random variables $X = \{X_1, \ldots, X_D\}$ when its constituent members are multiplied together. That is, for an element $(P_i : i \in \mathcal{V}) \in \mathcal{C}_\mathcal{G}$ the joint is $\prod_{i=1}^{D} P_i(X_i|X_{\mathrm{PA}_\mathcal{G}(i)})$. Hence $\mathcal{C}_\mathcal{G}$ induces a set of joint distributions $\mathcal{F}_\mathcal{G} = \{\prod_{i=1}^{D} P_i(X_i|X_{\mathrm{PA}_\mathcal{G}(i)}) : (P_i : i \in \mathcal{V})\}$ (Dhir et al., 2024). We also assume likelihood modularity (Geiger & Heckerman, 2002), that is for variable $X_i$ if $\mathrm{PA}_\mathcal{G}(i) = \mathrm{PA}_{\mathcal{G}'}(i)$ for $\mathcal{G} \neq \mathcal{G}'$, then $\mathcal{C}_{i|\mathrm{PA}_\mathcal{G}(i)} = \mathcal{C}_{i|\mathrm{PA}_{\mathcal{G}'}(i)}$. This means that for graphs where variables have the same parents, we assume the same set of conditional distributions.

### 2.2. Learning Causal Structure

Formally, the task is to recover a DAG $\mathcal{G}$ given a dataset of $N$ samples with $D$ variables, $\mathbf{X} \in \mathbb{R}^{N \times D}$. Without any interventional data, only the *Markov equivalence class* (MEC) of a DAG can be recovered with the faithfulness assumption (Pearl, 2009) and with infinite data (Appendix A). To recover a unique DAG, and differentiate *within* an MEC, we require additional assumptions.

### 2.3. Learning Causal Structure with Functional Restrictions

One way to recover a DAG is to impose a priori restrictions on the allowable conditional distributions $\mathcal{C}_{i|\mathrm{PA}_\mathcal{G}(i)}$. These restrictions are chosen so that the joint distributions implied by a causal model cannot be expressed by any other causal model— specifically, for $P \in \mathcal{F}_\mathcal{G}$ it holds that $P \notin \mathcal{F}_\mathcal{H}$ for any $\mathcal{H} \neq \mathcal{G}$ (Guyon et al., 2019, Ch. 2). Thus, if data is sampled from one of the causal models, the causal structure can be identified. As an example, additive noise models (ANM) (Hoyer et al., 2008) restrict all $\mathcal{C}_{i|\mathrm{PA}_\mathcal{G}(i)}$ to the form $f(X_{\mathrm{PA}_\mathcal{G}(i)}) + \epsilon$ for some arbitrary non-linear function $f$, and arbitrarily distributed $\epsilon$. If we consider a different graph $\mathcal{H}$ but the same ANM restrictions on $\mathcal{C}_{i|\mathrm{PA}_\mathcal{H}(i)}$, it is not possible to approximate the joint induced by the original set of conditional distributions $\mathcal{C}_\mathcal{G} = \prod_{i \in \mathcal{V}} \mathcal{C}_{i|\mathrm{PA}_\mathcal{G}(i)}$ using $\mathcal{C}_\mathcal{H} = \prod_{i \in \mathcal{V}} \mathcal{C}_{i|\mathrm{PA}_\mathcal{H}(i)}$. This allows the graph $\mathcal{G}$ to be identified. For a lot of these restrictions, the maximum likelihood score can identify the causal structure (Zhang et al., 2015; Peters & Bühlmann, 2014; Immer et al., 2023).

Methods relying on restricted model classes also restrict the datasets they can model. Specifically, assume that data was generated by a distribution $\Pi$, then it may be that $\Pi \notin \mathcal{F}_\mathcal{G}$ for all $\mathcal{G}$. Here, none of the causal models can approximate the true data-generating distribution and guarantees about causal identifiability no longer hold. That is, the models are misspecified. A solution would be to loosen the restrictions on $\mathcal{C}_{i|\mathrm{PA}_\mathcal{G}(i)}$ providing greater flexibility in approximating distributions. However, this could also allow causal models with different DAGs to express the true data-generating distribution, thereby losing the ability to recover the true DAG (Zhang et al., 2015, Lemma 1).

### 2.4. Learning Causal Structure with Bayesian Model Selection

Restricted function classes are required for identifiability within an MEC as it implies that, with infinite data, only one of the causal models can approximate the data distribution and thus achieve a higher score such as the maximum likelihood. Without restrictions, multiple causal structures can yield the same maximum likelihood score, making it impossible to distinguish between them. However, using such an

unrestricted causal model with a Bayesian model selection-based score can restore the ability to differentiate between causal structures. This is because, while multiple structures may share the same maximum likelihood score, they typically yield distinct scores under Bayesian model selection. This insight was demonstrated for the two-variable case in Dhir et al. (2024), building on prior efforts (Friedman & Nachman, 2000; Stegle et al., 2010). While this approach no longer guarantees strict identifiability and admits a small probability of error, the error probability can be empirically estimated and remains low with appropriate models (Dhir et al., 2024). The key advantage of this framework lies in its flexibility. By enabling the use of broader model classes, it mitigates the risk of misspecification, which is a common limitation where the assumptions and guarantees of restricted models fail (Section 2.3). Our work builds on these insights.

In Bayesian model selection, the evidence for a causal model $\mathcal{M}_\mathcal{G}$ (as defined in Section 2.1) is quantified by the posterior

$$P(\mathcal{M}_\mathcal{G}|X) \propto P(X|\mathcal{M}_\mathcal{G})P(\mathcal{M}_\mathcal{G}), \tag{1}$$

where $P(\mathcal{M}_\mathcal{G})$ is the prior over the causal model. As we assume no preference over causal direction between variables, our prior belief is equal over graphs in the same MEC. The posterior within an MEC is then dependent only on the term $P(X|\mathcal{M}_\mathcal{G})$, known as the *marginal likelihood*. Given that our task is to infer the single ground truth causal structure, we select the causal model $\mathcal{M}_\mathcal{G}^*$ based on the highest posterior probability (Kass & Raftery, 1995)

$$\mathcal{M}_\mathcal{G}^* = \underset{\mathcal{M}_\mathcal{G}}{\operatorname{argmax}} \, P(\mathcal{M}_\mathcal{G}|X). \tag{2}$$

To calculate the marginal likelihood, a prior (denoted $\pi$) over the distributions in the causal model must be specified. This defines a *Bayesian Causal Model*.

**Definition 2.2.** (Dhir et al., 2024) A *Bayesian Causal Model*, denoted as $(\mathcal{M}_\mathcal{G}, \pi)$, is defined as a causal model $\mathcal{M}_\mathcal{G}$ (Definition 2.1) with a prior distribution $\pi$ over $\mathcal{C}_\mathcal{G}$.

Each Bayesian causal model should encode the common and foundational principle that a change in one causal mechanism should not affect any others (Aldrich, 1989; Pearl, 2009). This can be formalised by the *independent causal mechanism* (ICM) assumption (Janzing & Schölkopf, 2010). The ICM assumption implies that, in the causal factorisation, knowledge about any distribution $P_i \in \mathcal{C}_{i|\text{PA}_\mathcal{G}(i)}$ should not inform any other $P_j \in \mathcal{C}_{j|\text{PA}_\mathcal{G}(j)}$ for $j \neq i$. In the Bayesian approach this implies defining factorised priors on each set of variable distributions — $\pi = \prod_{i \in \mathcal{V}} \pi_i$, where $\pi_i \in \mathcal{P}(\mathcal{C}_{i|\text{PA}_\mathcal{G}(i)})$ is the prior on the distributions for variable $X_i$, factorised according to the causal structure $\mathcal{G}$, and $\mathcal{P}$ is the set of all distributions over an object. This is the only constraint we place on the prior.

Multiple Bayesian causal models may achieve the same likelihood score given infinite data, but the prior and the encoded ICM assumption play a key role in yielding distinct marginal likelihood scores. The ICM assumption, which encodes the causal factorisation in the prior, typically does not hold in any other factorisation (Dhir et al., 2024). To see this, consider the model with graph $X \rightarrow Y$, with $\mathcal{C}_X$ parametrised by $\theta$ and $\mathcal{C}_{Y|X}$ parametrised by $\phi$. Here, placing a prior on $\mathcal{C}_{X \rightarrow Y}$ is equivalent to placing a prior on the parameters $\theta$ and $\phi$. ICM implies that the priors over $\theta, \phi$ factorise, which means the joint factorises as

$$\pi(\theta)\pi(\phi)P(X|\theta)P(Y|X, \phi). \tag{3}$$

We can find the reverse factorisation of the variables for the same model using Bayes' rule. Here, in general, it is not possible to parametrise $\mathcal{C}_Y$ and $\mathcal{C}_{X|Y}$ with independent parameters (Dhir et al., 2024)[2]

$$\text{Equation (3)} = \pi(\theta)\pi(\phi)P(Y|\theta, \phi)P(X|Y, \theta, \phi) \tag{4}$$
$$= \pi(\alpha, \beta)P(Y|\alpha)P(X|Y, \beta), \tag{5}$$

where $\alpha, \beta$ are some reparametrisations. Hence, if $\pi(\alpha, \beta) \neq \pi(\alpha)\pi(\beta)$, the model with the graph $X \rightarrow Y$ only satisfies the ICM assumption, as specified by factorised priors on the set of conditionals, in the causal factorisation of the joint $\{P_i(X)P_j(Y|X) : P_i \in \mathcal{C}_X, P_j \in \mathcal{C}_{Y|X}\}$.

Given a prior, the marginal likelihood is

$$P(X|\mathcal{M}_\mathcal{G}) = \int \prod_{i=1}^{D} \left( P_i(X_i|X_{\text{PA}_\mathcal{G}(i)})\pi_i(\mathrm{d}P_i) \right). \tag{6}$$

The marginal likelihood score has an automatic Occam's razor effect that balances model fit along with complexity (MacKay, 2003, Ch. 28). This means that although multiple causal models may have parameter settings that fit a given distribution, some models may provide a simpler explanation than others. Specifically, those whose prior, and encoded ICM assumption, align with properties of the data generating process. This allows for distinguishing between causal models even with loosening the restrictions discussed in Section 2.3.

Dhir et al. (2024) only consider Bayesian model selection for the two variable case. It is not clear how this approach will perform when the number of variables is increased, especially in contrast with competing methods. We study the multiple variable case where we address the following challenges: First, we use the theory in Dhir et al. (2024) to clearly show that Bayesian model selection can be applied to causal discovery with multiple variables. Secondly, as the number of variables increases, the number of possible DAGs

---

[2]Exceptions exist, notably normalised linear Gaussian models which are known to be unidentifiable.

grows super-exponentially, making direct comparison of marginal likelihoods across all DAGs infeasible. To address this, we propose a scalable approach for applying Bayesian model selection to multiple-variable causal discovery.

## 3. Distinguishability with Multiple Variables

We use theory from Dhir et al. (2024) to show that in the multiple variable case, Bayesian causal models can find the correct DAG using only priors with the ICM condition, rather than hard model restrictions. We proceed by first showing that Bayesian causal models can be identified up to an MEC. Then, we show that Bayesian causal models can differentiate within an MEC, without functional restrictions, but with some probability of error. We only make remarks on key results in this section, but the full set of assumptions, theorems, and proofs is in Appendix B. Note that these definitions hold in the population setting.

First, we define the probability of making an error, given datasets from the chosen Bayesian causal model.

**Definition 3.1.** (Dhir et al., 2024) Given a score $\mathcal{S}$, such as the marginal likelihood, we define the probability of error for a model $\mathcal{M}_{\mathcal{G}}$ as

$$P(E|\mathcal{M}_{\mathcal{G}}) = \int_{\mathcal{R}} p(X|\mathcal{M}_{\mathcal{G}})dX, \qquad (7)$$

where $\mathcal{R} = \{X : \mathcal{S}(X|\mathcal{M}_{\mathcal{H}}) > \mathcal{S}(X|\mathcal{M}_{\mathcal{G}})$ for $\mathcal{H} \neq \mathcal{G}\}$, and $\mathcal{S}(X|\mathcal{M}_{\mathcal{G}})$ is the score achieved by model $\mathcal{M}_{\mathcal{G}}$ on dataset $X$.

That is, given datasets $X$ from a Bayesian causal model with DAG $\mathcal{G}$, the integral over datasets where the wrong causal model is chosen by the score. For Bayesian causal models, this quantifies the overlap in the posteriors. Note that, similar to previous work, this definition assumes that the data is sampled from one of the models under consideration. The probability of error can thus be estimated empirically for Bayesian causal models by sampling a graph, then sampling datasets from a chosen Bayesian causal model, and inferring the causal graph using the chosen score (Dhir et al., 2024). An *identifiable* model implies a probability of error of zero in the infinite data limit (Dhir et al., 2024; Guyon et al., 2019). In contrast, an *unidentifiable* model will have a probability of error equal to that of a uniformly random chosen graph. Similarly, we say a model is *distinguishable* if the probability of error is less than that of a uniformly chosen random graph.

*Remark* 3.2. With the faithfulness assumption, Bayesian model selection can identify a Bayesian causal model up to the MEC of its graph (Theorem B.3).

Although this is well known for certain parametric models (Heckerman, 1995; Chickering, 2002), we show that it is true for the general non-parametric case.

The advantage of the Bayesian approach is apparent when considering cases where the underlying causal model is unidentifiable with maximum likelihood. This is the case for graphs within an MEC, where functional restrictions have not been made.

*Remark* 3.3. Suppose that the ICM assumption only holds in the causal factorisation of the Bayesian causal model. Bayesian model selection can then distinguish within an MEC (Theorem B.7).

Thus, without restrictions and if the ICM condition only holds in the causal factorisation, Bayesian model selection can be used to distinguish Bayesian causal models, in that the probability of error is lower than a uniformly random chosen graph. The model class we choose only satisfies the ICM condition in the causal factorisation (Dhir et al., 2024, Appendix D.3). The probability of error for our chosen model class was shown to be very low for the two variable case in Dhir et al. (2024, Section 4.3). We estimate the probability of error for the multivariate case and show that it is low in Section 6.1.

As is common with identifiability results, the probability of error estimate assumes that the data is generated from one of the candidate causal models. However, this assumption may not hold in practice. Dhir et al. (2024, Section 4.4) demonstrate that the model's estimated probability of error and the true probability of error (from the actual data-generating process) are bounded by the total variation distance between the model and the true distribution. Consequently, mild violations of model assumptions, such as the prior, can still yield accurate error probability estimates.

## 4. CGP-CDE: Causal Gaussian Process Conditional Density Estimators

Although the posterior can allow for more accurate causal discovery with fewer restrictions, computing and comparing the posterior of every causal structure becomes intractable when the number of variables is increased. We propose a method that scales up Bayesian model selection based causal discovery by using the insights of Zheng et al. (2018) (Appendix A.2). We first describe the model that we use, then we show how the model can continuously parameterise the space of graphs. This is followed by a loss function that maximises the posterior probability and ensures that the final graph is a DAG.

### 4.1. The GP-CDE Model

The Bayesian model selection approach to causal structure learning described in Section 2.4 requires defining distributions and priors for a causal model. To take advantage of the Bayesian framework, we wish to use a model for each $\mathcal{C}_{i|\mathrm{PA}_{\mathcal{G}}(i)}$ that is more flexible than previous attempts. For

this, we use a version of the *Gaussian process conditional density estimator* (GP-CDE) model (Dutordoir et al., 2018). Here, $\mathcal{C}_{i|\mathrm{PA}_{\mathcal{G}}(i)}$ is a class of nonparametric functions and the prior over the conditionals is a Gaussian process prior.

The input to the estimator for each variable is determined by the causal graph of the causal model. For a given causal DAG $\mathcal{G}$, the joint $p(\mathbf{X}, \mathbf{f}, \mathbf{W}|\mathbf{\Lambda}, \phi, \mathcal{M}_{\mathcal{G}})$ of our model is

$$\prod_{i=1}^{D} \prod_{n=1}^{N} p(x_{ni}|\mathbf{f}_i, \phi_i)p(\mathbf{f}_i|\mathbf{X}_{\mathrm{PA}_{\mathcal{G}}(i)}, \mathbf{\Lambda}_i, w_{ni})p(w_{ni}), \quad (8)$$

where $\mathbf{W} \in \mathbb{R}^{N \times D}$ are the latent variables, $\phi \in \mathbb{R}^{D}$ are the likelihood variances, $\mathbf{\Lambda}_i$ are the kernel hyperparameters, and $\mathcal{M}_{\mathcal{G}}$ denotes the causal model with DAG $\mathcal{G}$. Each variable $\mathbf{x}_i$ is modelled as the output of a function $\mathbf{f}_i$, with the parents of the variable in $\mathcal{G}$, $\mathbf{X}_{\mathrm{PA}_{\mathcal{G}}(i)}$, and a latent variable $\mathbf{w}_i$ being the inputs. The inclusion of $\mathbf{w}_i \sim \mathcal{N}(\mathbf{0}, \mathbf{I})$ allows for each variable to be described with non-Gaussian and heteroscedastic noise, greatly increasing the expressivity of the model (Dutordoir et al., 2018)[3]. The term $p(\mathbf{f}_i|\mathbf{X}_{\mathrm{PA}_{\mathcal{G}}(i)}, \mathbf{\Lambda}_i, w_{ni})$ is the Gaussian process prior (Rasmussen, 2003), and equal to

$$\mathcal{N}\big(\mathbf{0}, K_{\mathbf{\Lambda}_i}\big((\mathbf{X}_{\mathrm{PA}_{\mathcal{G}}(i)}, \mathbf{w}_i), (\mathbf{X}_{\mathrm{PA}_{\mathcal{G}}(i)}, \mathbf{w}_i)'\big)\big), \quad (9)$$

where $K(\cdot, \cdot)$ is a chosen kernel matrix parameterised by hyperparameters $\mathbf{\Lambda}_i$.

The above approach requires defining a separate model for each causal DAG. Next, we show how to interpret the hyperparameters of this model as an adjacency matrix. This will allow for continuously parameterising the space of graphs.

### 4.2. Continuous Relaxation: The CGP-CDE Model

The key insight to continuously parameterise graphs (not DAGs) in the above model is that the hyperparameters control dependence between variables (Williams & Rasmussen, 1995). In our Gaussian process prior for a variable $X_i$, we use kernels with different hyperparameters $\Lambda_{ij}$ for each dimension $j$. In general, kernels of this form factorise as $K_{\mathbf{\Lambda}_i}((\mathbf{X}, \mathbf{w}_i), (\mathbf{X}, \mathbf{w}_i)') = K_{\Lambda_{ii}}(\mathbf{w}_i, \mathbf{w}_i') \prod_{j \neq i} K_{\Lambda_{ij}}(\mathbf{x}_j, \mathbf{x}_j')$. Certain hyperparameters of the kernels can then be used to control the variability of the function with respect to specific inputs. We denote these hyperparameters as $\boldsymbol{\theta}$ with $\boldsymbol{\theta} \subset \mathbf{\Lambda}$ and all other hyperparameters as $\boldsymbol{\sigma} = \mathbf{\Lambda} \setminus \boldsymbol{\theta}$. Specifically

$$\theta_{ij} = 0 \implies \frac{\partial f_i}{\partial X_j} = 0.$$

Hence, if the kernel hyperparameter value for $\theta_{ij}$ is near zero, the function for variable $X_i$ is constant with respect to

---

[3]Our model can be thought of as a Bayesian conditional variational autoencoder.

variable $X_j$. This implies an adjacency matrix

$$A_{ij} = \begin{cases} \theta_{ij} & \text{if } j \neq i, \\ 0 & \text{otherwise.} \end{cases} \quad (10)$$

A value of $\theta_{ij} = 0$ implies that changing $X_j$ does not change $X_i$ and results in an absence of the edge $X_j \to X_i$ in $\mathbf{A}$.

The Gaussian process prior $p(\mathbf{f}_i|\mathbf{X}_{\mathrm{PA}_{\mathcal{G}\mathbf{A}}(i)}, \boldsymbol{\sigma}_i, \boldsymbol{\theta}_i, \mathbf{w}_i)$ is then parameterised as follows

$$\mathcal{N}(\mathbf{0}, K_{\boldsymbol{\sigma}_i, \boldsymbol{\theta}_i}((\mathbf{X}_{\neg i}, \mathbf{w}_i), (\mathbf{X}_{\neg i}, \mathbf{w}_i)')), \quad (11)$$

where $\mathrm{PA}_{\mathcal{G}\mathbf{A}}(i)$ denotes the parents of $X_i$ in $\mathcal{G}^{\mathbf{A}}$ (the graph implied by the adjacency $\mathbf{A}$), and $\mathbf{X}_{\neg i}$ denotes all variables except $X_i$. Thus, for a variable $X_i$, all the other variables are inputs, and the values of the hyperparameters $\boldsymbol{\theta}_i$ control the dependence of the rest of the variables $\mathbf{X}_{\neg i}$ on $X_i$, and hence define the parents of $X_i$. The hyperparameters in the above model thus continuously parameterise the space of graphs. The exact kernel we use is in Appendix C.1.

### 4.3. Priors on Graphs

As discussed in Eggeling et al. (2019), the lack of a prior on graphs leads to a higher weight on denser graphs due to the larger number of dense graphs versus sparse graphs. Hence, to even out this effect, we place a prior on graphs. By introducing priors on the hyperparameters $\boldsymbol{\theta}$ in our model, we can effectively encode priors on the graph in Equation (10). We thus place a Gamma prior with parameters chosen to prefer small values of $\boldsymbol{\theta}$, $P(\boldsymbol{\theta}) = \mathrm{Gamma}(\eta, \beta)$. Note that our prior is symmetric between graphs within the same MEC. That is, we do not impose a preference over the causal direction between variables, only over the number of edges in the graph. Other priors on graphs could be considered (Eggeling et al., 2019), but we leave that for future work.

### 4.4. Score

As stated in Section 2.4, our decision rule is to pick the causal graph with the highest posterior probability. For this, we need to calculate the log marginal likelihood for every variable in our model (Equation (6)). To find the log marginal likelihood, we must integrate over the priors of $\mathbf{f}$ and $\mathbf{W}$, $\log p(\mathbf{x}_i|\mathbf{X}_{\mathrm{PA}_{\mathcal{G}\mathbf{A}}(i)}, \boldsymbol{\sigma}_i, \boldsymbol{\theta}_i, \phi_i)$ is given as

$$\log \int \int p(\mathbf{x}_i|\mathbf{f}_i, \phi_i)p(\mathbf{f}_i|\mathbf{X}_{\mathrm{PA}_{\mathcal{G}\mathbf{A}}(i)}, \mathbf{w}_i, \boldsymbol{\sigma}_i, \boldsymbol{\theta}_i)p(\mathbf{w}_i)d\mathbf{w}_i d\mathbf{f}_i.$$

Calculating this marginal likelihood is analytically intractable due to the non-linear dependence of $\mathbf{f}_i$ on the latent term $\mathbf{w}_i$. We thus use variational inference, with variational posteriors $q$, to optimise a tractable lower bound to the marginal likelihood (details and derivation

as well as handling of other hyperparameters are in Appendix C.2). We denote the full lower bound for all variables as $\mathcal{L}_{\text{ELBO}}(q, \boldsymbol{\theta}, \boldsymbol{\sigma}, \phi)$. The optimisation problem is

$$\max_{\boldsymbol{\theta}, \boldsymbol{\sigma}, \phi, q} \mathcal{L}_{\text{ELBO}}(q, \boldsymbol{\theta}, \boldsymbol{\sigma}, \phi) + \log p(\boldsymbol{\theta}) \text{ s.t. } \mathcal{G}^{\mathbf{A}_{\boldsymbol{\theta}}} \in \text{DAGs}.$$

Maximising $\boldsymbol{\sigma}, \phi$, and $q$ gives an accurate estimate of the marginal likelihood while maximising $\boldsymbol{\theta}$ finds the graph that maximises the posterior probability.

To enforce the condition that $\mathcal{G}^{\mathbf{A}_{\boldsymbol{\theta}}} \in$ DAGs we use the spectral acyclicity constraint method (Lee et al., 2019)

$$h(\mathbf{A}_{\boldsymbol{\theta}}) = |\lambda_d(\mathbf{A}_{\boldsymbol{\theta}})|, \tag{12}$$

where $\lambda_d(\mathbf{A}_{\boldsymbol{\theta}})$ denotes the largest eigenvalue of $\mathbf{A}_{\boldsymbol{\theta}}$, which has been shown to be more stable and scalable than other acyclic constraints (Nazaret et al., 2024). Following Nazaret et al. (2024), a penalty method is used to allow for continuous optimisation, making the final loss

$$\mathcal{L}_{\text{ELBO}}(q, \boldsymbol{\theta}, \boldsymbol{\sigma}, \phi) + \log p(\boldsymbol{\theta}) - \gamma_t h(\mathbf{A}_{\boldsymbol{\theta}}), \tag{13}$$

where the weighting $\gamma_t$ is increased by $\rho$ at each epoch. A solution $\mathbf{A}^*$ is found when $h(\mathbf{A}^*) < \tau$ for some $\tau > 0$. See Appendix A.2 for more details.

### 4.5. Computational Cost

The cost of computing the loss for $N$ samples in a Gaussian Process is usually $\mathcal{O}(N^3)$. We use the uncollapsed inducing point formulation that introduces $M < N$ inducing points allowing for mini-batching (Appendix C.2)(Hensman et al., 2013). This changes the cost to $\mathcal{O}(M^3)$. As we have $D$ variables, the total cost is $\mathcal{O}(DM^3)$. The cost of computing the acyclicity regulariser is $\mathcal{O}(D^2)$. Hence, the cost is dominated by $M$. The choice of $M$ depends on the problem, with more inducing points leading to better approximations (Bauer et al., 2016). However, it is possible to use $\mathcal{L}_{\text{ELBO}}$ to find the optimal number of inducing points (Burt et al., 2020). Gaussian processes, and Bayesian methods in general, can be more computationally expensive than their non-Bayesian counterparts. However, Bayesian model selection-based causal discovery provides key advantages, including the ability to use more flexible model classes.

## 5. Related Work

Certain approaches only use information about independences between variables to discover the causal structure, recovering only an MEC, and not the whole DAG (Spirtes et al., 2000; 1995; Chickering, 2002; Huang et al., 2018). Identifying a unique DAG requires additional assumptions, typically involving restrictions on model classes or access to interventional data (Lippe et al., 2021; Ke et al., 2022;

Brouillard et al., 2020; Ke et al., 2020). Since we assume access only to observational data, we focus on the former.

Continuous optimisation methods cast causal discovery as a single unconstrained optimisation problem (Zheng et al., 2018). NOTEARS (Zheng et al., 2018) first introduced a regulariser for learning acyclic causal graphs in linear causal models. Follow-up work extended these methods to more flexible models such as neural networks, or graph neural networks (Lachapelle et al., 2019; Yu et al., 2019; Zheng et al., 2020). At the same time, works such as NOBEARS (Lee et al., 2019) and SDCD (Nazaret et al., 2024) have worked on improving the scalability of the acyclic regularisers. However, the above methods rely on restricted models to recover a DAG, which can limit performance when assumptions are not met (Section 2.3). When these models are not restricted, their guarantees no longer hold.

Other methods first search for node orderings and then prune to ensure a DAG structure. CAM (Bühlmann et al., 2014) uses sparse regression to learn node neighbourhoods and searches for orderings that maximise a score. SCORE (Rolland et al., 2022) learns the ordering based on the score function of ANM models with Gaussian noise and then prunes using sparse regression. NOGAM (Montagna et al., 2023) extends SCORE to arbitrary noise distributions. The above methods specifically rely on the ANM assumption.

The Bayesian framework has previously been used for causal discovery but has typically focused on computing a posterior over causal structures to quantify uncertainty (Friedman & Koller, 2003; Heckerman et al., 2006). Geiger & Heckerman (2002) use Bayesian linear Gaussian models but construct their model such that graphs in the same Markov equivalence class receive the same score. Similarly, Cundy et al. (2021) also only consider posteriors over linear Gaussian models. DiBS (Lorch et al., 2021) assumes Bayesian ANMs and parametrises the space of graphs using latent variables. Annadani et al. (2023) also use sampling methods to sample from the posterior over ANM causal models, but use a permutation based parametrisation of the graph (Charpentier et al., 2021; Yu et al., 2021).

Previous Bayesian methods have also relied on the restrictive assumptions common in likelihood based methods. In contrast, our work demonstrates that the advantage of Bayesian model selection is the allowance of fewer restrictions compared to likelihood-based methods (Stegle et al., 2010; Dhir et al., 2024; Cooper & Herskovits, 1992). This is strongly related to the idea of using independence of causal mechanisms (and the Kolmogorov complexity of causal models) to distinguish causal models (Janzing & Schölkopf, 2010) (see Dhir et al. (2024, Appendix B)). While previous studies demonstrated this with two variables, we show its feasibility and advantages in the important multi-variable regime.

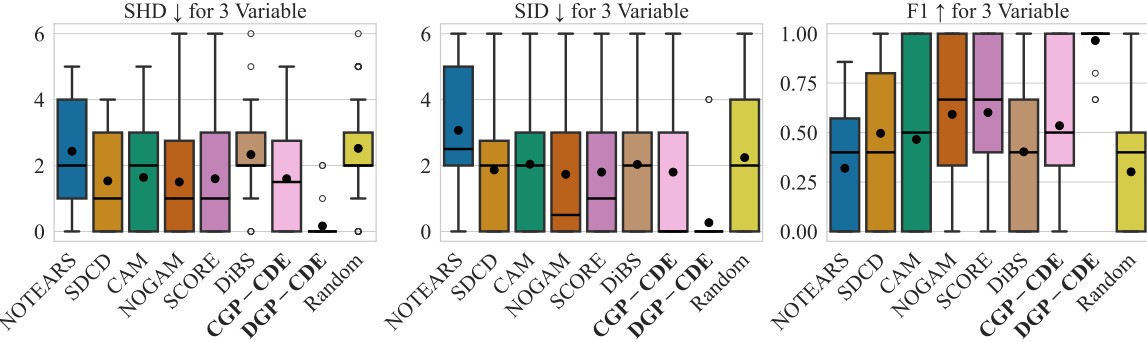

Figure 1: Boxplot of metrics for the three variable dataset. The plots show the distribution of the metrics across the different datasets for each experiment. The centre line is the median, the black dot is the mean, and the white dots represent outliers. Lower is better for SHD and SID, and higher is better for F1.

## 6. Experiments

We compare our approach to various baselines that perform multivariate causal discovery to recover a DAG. We test our method on synthetic data generated from our model (Section 6.1), and data not generated from our model (Section 6.2). Then we test our model on a common semi-synthetic benchmark (Section 6.3). Additional experiments and results are in Appendix I. See Appendix C.6 for full details of the CGP-CDE implementation. Code can be found at: `https://github.com/Anish144/ContinuousBMSStructureLearning.git`.

It was recently found that causal discovery methods can be sensitive to variance information that is an artefact of synthetic causal datasets (Reisach et al., 2021; Ormaniec et al., 2024). Hence we standardise every variable, including during data generation (see Appendix E).

**Baselines:** Details for all baselines are in Appendix G. We compare against non-Bayesian counterparts that use acyclicity regularisers, such as NOTEARS (Zheng et al., 2018), and SDCD (Nazaret et al., 2024). SDCD uses the same spectral acyclicity regulariser as the CGP-CDE, the only difference being that the model uses a neural network and a likelihood score to learn the data distribution (Lachapelle et al., 2019). Thus, comparison against this method provides direct evidence for the usefulness of the Bayesian approach. We also compare against methods that first estimate a topological order of the graph and then prune to get the final graph - SCORE (Rolland et al., 2022), NOGAM (Montagna et al., 2023), and CAM (Bühlmann et al., 2014). These all assume ANM. Finally, we compare against the MAP estimate of the Bayesian model DiBS. Other Bayesian methods are sample based and hence do not allow for easy MAP calculation.

**Metrics:** We use the following metrics to compare the methods. **SHD**: The structural Hamming distance is the Hamming distance between the predicted adjacency and the ground truth adjacency matrix. **SID**: The structural interventional distance counts the number of interventional distributions that are incorrect if the predicted graph is used to form the parent adjustment set instead of the ground truth graph (Peters & Bühlmann, 2015). **F1**: The F1 score is the harmonic mean of the precision and recall, where an edge is considered the positive class.

### 6.1. Synthetic 3 Variables

Using more flexible estimators may introduce some probability of error (Section 3). In the two variable case, Dhir et al. (2024) showed that the probability of error for our model class is low; we show that this is true for the multivariate case as well. To estimate the error only due to overlap in posteriors of different causal models, we perform discrete model comparison (labelled DGP-CDE, see Appendix D.1 for more details). This involves enumerating and computing the posterior of every causal model. We study the 3 variable case, where it is possible to enumerate every possible causal structure, 25 in total. Here, as required by the probability of error, data is generated from our GP-CDE model (details in Appendix F). We generate five datasets of 1000 samples for each of the six graphs that are unique up to a permutation of the variables.

The results for the 3 variable case can be seen in Figure 1. Here, we can see that the discrete comparison DGP-CDE achieves very good performance. This shows that in the GP-CDE model, the probability of error in the multivariable case is low and that our inference does not contribute greatly to errors. Thus, if the GP-CDE model is a good descriptor of the dataset at hand, we can expect the MAP value to identify the true causal direction with high probability. We also compare our continuous relaxation model against the discrete model that enumerates each graph type. In Figure 1, the performance of CGP-CDE shows that we can expect a higher error due to optimisation of the continuous relaxation

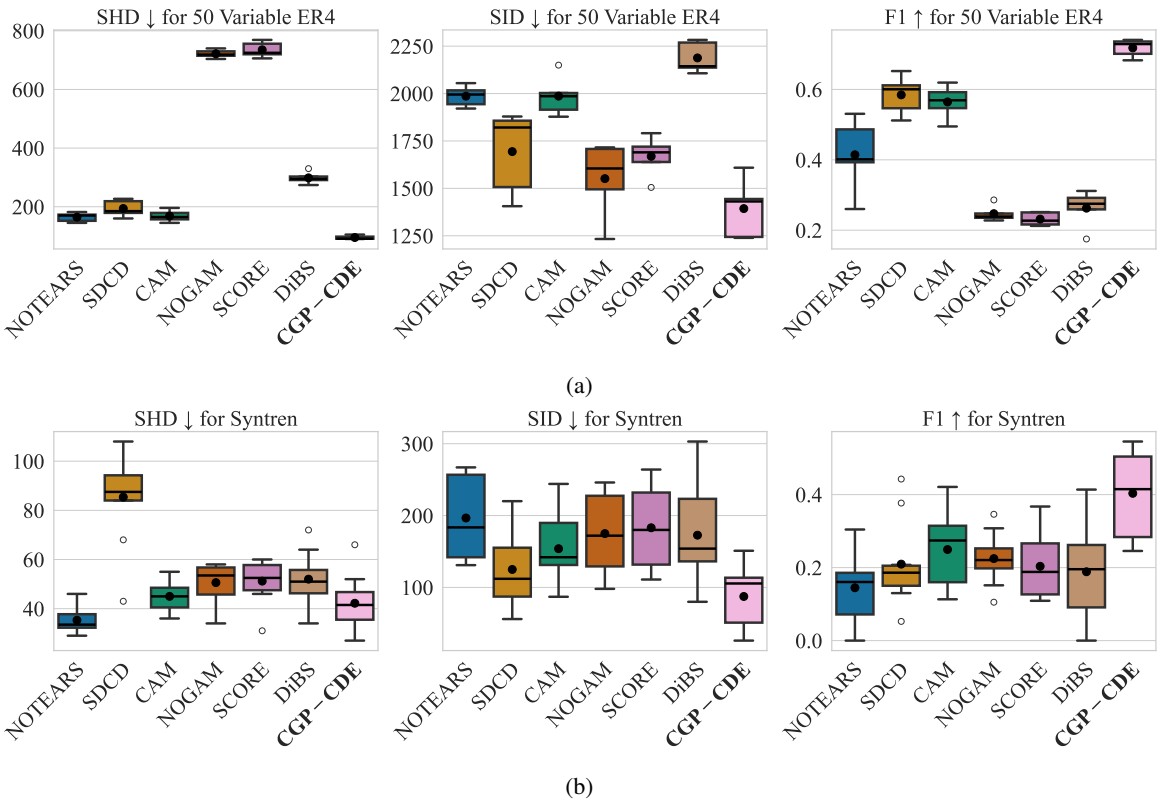

Figure 2: Box plots of metrics for different graph types: (a) Erdos-Renyi (ER) graphs with four expected edges per variable, (b) Syntren data. Black dot is the average metric. Lower is better for SHD and SID, higher is better for F1.

in the CGP-CDE. However, the CGP-CDE still performs competitively compared to other baselines.

### 6.2. Synthetic Higher Variable

Next, we analyse the performance on a higher number of variables with varying graph density. Here, performing discrete model comparison is too costly as the number of DAGs is prohibitively large, so we use our continuous relaxation, labelled CGP-CDE. We generate five datasets for each graph size 50 using Erdos-Rényi (ER) and scale-free (SF) (Barabási & Albert, 1999) graph generation schemes, with expected total edge counts of 1D and 4D, respectively. In ER graphs, each edge is generated independently, while SF graphs promote a few nodes with a high number of edges. To test our claim that Bayesian model selection allows for more flexible models, we ensure that the data is not generated from our model. The data is generated from randomly initialised neural networks with noise included as an input. Performing well in such cases showcases the advantage of the added flexibility in the Bayesian approach. Data generation details are in Appendix F. To show that our method also performs well with a lower number of variables and samples, we show results for a graph size of 20 in Appendix I.3 and Appendix I.2. We also perform an ablation study with ANM

datasets in Appendix I.1.

The full results are in Appendix I, and we show the 50 variable ER4 results in Figure 2a. Compared to the DGP-CDE results of Section 6.1, we can expect a higher number of errors due to the continuous relaxation and the shift in the data generation process. Nevertheless, the CGP-CDE outperforms all other baselines, especially in terms of SHD and F1 scores, and the increase in performance compared to the baselines is greater for these larger graphs. While both SDCD and CGP-CDE employ flexible function approximators and the same acyclic regulariser, the superior performance of CGP-CDE suggests that Bayesian methods are more effective at distinguishing causal structures. In contrast, the poor performance of DiBS (MAP estimation) can be attributed to poor inference in Bayesian neural networks.

### 6.3. Syntren

Syntren is a gene-regulatory network simulator that generates gene expression data from real biological graphs. There are 10 datasets of 20 nodes with 500 samples each. The results on Syntren can be seen in Figure 2b. Here, the CGP-CDE outperforms other methods, especially in terms of the SID and F1 scores.

# 7. Conclusion

In this work, we have shown why Bayesian model selection works, and that it is a highly effective approach for causal discovery in multivariable settings. Then, we propose a scalable model that outperforms existing Bayesian and non-Bayesian methods across multiple settings. The consistent performance of our approach highlights the advantages of framing causal discovery as a Bayesian model selection problem. We believe this perspective addresses a critical limitation in causal discovery—namely, the inability to use flexible models, paving the way for more practical and widely applicable causal inference in real-world scenarios.

# Impact Statement

This paper presents work whose goal is to advance the field of Machine Learning. There are many potential societal consequences of our work, none of which we feel must be specifically highlighted here.

# Acknowledgements

R.S. was supported by the Wellcome Trust [222836/Z/21/Z]. B.G. acknowledges the support of the UKRI AI programme, and the Engineering and Physical Sciences Research Council, for CHAI - EPSRC Causality in Healthcare AI Hub (grant no. EP/Y028856/1). A.K. is supported by UKRI (grant number EP/S023356/1), as part of the UKRI Centre for Doctoral Training in Safe and Trusted AI. A.D. acknowledges the support of G-Research's March 2025 grant.

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

# A. Additional Background

## A.1. Causal Model Basics

The causal model satisfies the *Independent Causal Mechanism* (ICM) assumption by construction (Janzing & Schölkopf, 2010). Given a graph $\mathcal{G}$, this means that changing a term in the causal factorisation of the joint $\prod_{i=1}^{D} P_i(X_i | X_{\text{PA}_{\mathcal{G}}(i)})$ does not change any of the other terms. This is achieved by assuming that the choice of conditional distributions for each variable $X_i$ from $\mathcal{C}_{i|\text{PA}_{\mathcal{G}}(i)}$ is independent of the choice for other variables $X_j$ from $\mathcal{C}_{j|\text{PA}_{\mathcal{G}}(j)}$ with $j \neq i$. This property is assumed to hold in the causal factorisation but does not necessarily hold in any other factorisation of the joint (Peters et al., 2017). This can be seen in the two variable system with graph $X \to Y$. If the terms in $P(X)P(Y|X)$ are chosen independently, the choice of the terms in $P(Y)P(X|Y)$ are tied to each other. Changing $P(X)$ or $P(Y|X)$ in general will require changes in both $P(Y)P(X|Y)$ to maintain equality.

As variables are dependent only on their parents in a graph by construction, a causal model satisfies the *Markov* assumption. This states that d-separations in $\mathcal{G}$ between variables implies conditional independence between the same variables (Pearl, 2009). We also assume *faithfulness*: any conditional independences between variables are represented by d-separations in $\mathcal{G}$ (Pearl, 2009). Together, faithfulness and Markov assumptions imply a one-to-one relationship between independences in the distributions and d-separations in the graph (Peters et al., 2017). These assumptions are enough to recover a causal structure from data up to a Markov equivalence class (MEC), that is, the class of causal structures that have the same d-separations.

## A.2. Continuous Optimisation for Learning Causal Structure

Given a score (such as the likelihood or marginal likelihood) $\mathcal{S}$, causal structure learning can be defined as an optimisation problem over graphs (Zheng et al., 2018)

$$\mathcal{G}^* = \underset{\mathcal{G}}{\arg\max}\, \mathcal{S}(\mathcal{G}) \text{ such that } \mathcal{G} \in \text{ DAGs.} \tag{14}$$

That is, finding the graph that optimises the score under the constraint that the graph must be a valid DAG. This converts causal structure learning into a single optimisation problem, tackling the scalability issue. However, ensuring that the graph is acyclic is non-trivial with gradient based methods. Zheng et al. (2018) first solve this by encoding a graph as a weighted adjacency matrix $\mathbf{A} \in \mathbb{R}_{\geq 0}^{D \times D}$ and using a measure of acyclicity $h(\mathbf{A})$ as a constraint. The function $h : \mathbf{A} \to \mathbb{R}_{\geq 0}$ is a measure of the number of weighted directed walks that allow for returning to a starting node. Hence, $h(\mathbf{A}) = 0$ implies that there are no directed cycles in the adjacency $\mathbf{A}$ and that the graph implied by $\mathbf{A}$ is a valid DAG.

Numerous variations on this approach have been proposed to improve the scalability and stability of optimisation (Lee et al., 2019; Bello et al., 2022; Wei et al., 2020; Nazaret et al., 2024). We use the spectral acyclicity constraint method in Equation (12), which is equivalent to the largest eigenvalue magnitude of $\mathbf{A}$, due to its stability and scalability (Lee et al., 2019; Nazaret et al., 2024). This constraint uses the fact that $\mathbf{A}$ is acyclic if and only if all its eigenvalues are zero (Cvetković et al., 1980).

The gradients of the spectral acyclicity constraint can be easily calculated as

$$\nabla h(\mathbf{A}) = \frac{v_d u_d^T}{v_d^T u_d}, \tag{15}$$

where $u_d$ and $v_d$ are the right and left eigenvectors associated with $\lambda_d(\mathbf{A})$ (Magnus, 1985). We follow the example of Nazaret et al. (2024) and use the power iteration method to estimate these eigenvectors in $\mathcal{O}(d^2)$ rather than finding them exactly, which costs $\mathcal{O}(d^3)$.

Equation (14) can now be written as an optimisation problem with the differentiable constraint $h(\mathbf{A}) = 0$. Nazaret et al. (2024) use a penalty method, writing the objective as:

$$\underset{\mathbf{A}}{\arg\max}\, \mathcal{S}(\mathcal{G}^{\mathbf{A}}) - \gamma_t h(\mathbf{A}), \tag{16}$$

where the weighting $\gamma_t$ of the penalty $h(\mathbf{A})$ is increased by $\rho$ each epoch. A solution $\mathbf{A}^*$ is found when $h(\mathbf{A}^*) < \tau$ for some convergence threshold $\tau > 0$.

# B. Proof of Distinguishability with Multiple Variables

Bayesian model selection uses the marginal likelihood (equivalent to the posterior over models) to distinguish between Bayesian causal models with different causal graphs. Thus, following Dhir et al. (2024), we will provide conditions such that two Bayesian causal models will have different distributions over marginal likelihood values. This implies that there will always exist datasets such that the marginal likelihood values of two Bayesian causal models differ. We will do this by defining two Bayesian causal models as Bayesian distribution-equivalent if they have the same distribution over marginal likelihood values, and then provide conditions such that two Bayesian causal models cannot be Bayesian distribution-equivalent. Note that this can be viewed as a version of observation equivalence (Pearl, 2009), but for Bayesian causal models. If models are not observationally equivalent, they can be learnt from observational data (for example, by using methods such as PC (Spirtes et al., 2000)). A similar reasoning holds for Bayesian distribution equivalence.

We split the proof into two parts, first, we will show that Bayesian model selection can effectively identify up to a Markov equivalence class. This is known for certain classes of models, for example, curved exponential models (Chickering, 2002), but it has not been shown for the general non-parametric case. Second, we will provide conditions such that two Bayesian causal models within the same Markov equivalence class are not Bayesian distribution-equivalent. Here, there may be some probability of error depending on the overlap of the posteriors that can be computed empirically. This second result is implied by the result in Dhir et al. (2024, Proposition 4.7).

We define Bayesian distribution-equivalent models. These are the Bayesian counterparts of models that are unidentifiable. That is, given any dataset, these models will achieve the same marginal likelihood score.

**Definition B.1.** (Dhir et al., 2024, Definition 4.4) Given two Bayesian causal models $(\mathcal{M}_\mathcal{G}, \pi_\mathcal{G})$, $(\mathcal{M}_\mathcal{H}, \pi_\mathcal{H})$, say they are **Bayesian distribution-equivalent** if $P(\cdot \mid \mathcal{M}_\mathcal{G}) = P(\cdot \mid \mathcal{M}_\mathcal{H})$, i.e. for all $N \in \mathbf{N}$, and for all $(\mathbf{x}^N, \mathbf{y}^N) \in (\mathcal{X} \times \mathcal{Y})^N$, it holds that $p(\mathbf{x}^N, \mathbf{y}^N \mid \mathcal{M}_\mathcal{G}) = p(\mathbf{x}^N, \mathbf{y}^N \mid \mathcal{M}_\mathcal{H})$.

We proceed by showing that two Bayesian causal models with graphs in different MECs cannot be Bayesian distribution equivalent. Further, if the estimate of the Bayes factor (Kass & Raftery, 1995) (ratios of marginal likelihood) is consistent, it can identify the correct graph.

**Lemma B.2.** *Assume faithfulness and two Bayesian causal models $(\mathcal{M}_\mathcal{G}, \pi_\mathcal{G})$, and $(\mathcal{M}_\mathcal{H}, \pi_\mathcal{H})$. If $\mathcal{G}$ and $\mathcal{H}$ are in separate Markov equivalence classes (MEC), then $\mathcal{F}_\mathcal{G} \cap \mathcal{F}_\mathcal{H} = \emptyset$, where $\mathcal{F}_\mathcal{G}$ and $\mathcal{F}_\mathcal{H}$ are the sets of all joints expressible by $\mathcal{M}_\mathcal{G}$ and $\mathcal{M}_\mathcal{H}$ respectively.*

*Proof.* The Markov and Faithfulness assumptions together means that $X_i \perp\!\!\!\perp_P X_j | X_k \iff X_i \perp\!\!\!\perp_\mathcal{G} X_j | X_k$, where $\perp\!\!\!\perp_P$ denotes distributional independence, and $\perp\!\!\!\perp_\mathcal{G}$ signifies d-separation in graph $\mathcal{G}$. Hence, as $\mathcal{G}$ and $\mathcal{H}$ are in separate MECs, there exists some $X_i \perp\!\!\!\perp_\mathcal{G} X_j | X_k$ such that $X_i \not\perp\!\!\!\perp_\mathcal{H} X_j | X_k$. This implies that for every $P \in \mathcal{F}_\mathcal{G}$, we have that $X_i \perp\!\!\!\perp_P X_j | X_k$, and for every $Q \in \mathcal{F}_\mathcal{H}$ we have that $X_i \not\perp\!\!\!\perp_Q X_j | X_k$. Hence $P \notin \mathcal{F}_\mathcal{H}$. $\square$

**Theorem B.3.** *Assume faithfulness and two Bayesian causal models $(\mathcal{M}_\mathcal{G}, \pi_\mathcal{G})$, and $(\mathcal{M}_\mathcal{H}, \pi_\mathcal{H})$. If $\mathcal{G}$ and $\mathcal{H}$ are in separate Markov equivalence classes (MEC), then they are not Bayesian distribution equivalent for any choices of priors $\pi_\mathcal{G}, \pi_\mathcal{H}$. Assume $N$ samples are generated from one of the models. If the estimate $\frac{P(X^N | \mathcal{M}_\mathcal{G})}{P(X^N | \mathcal{M}_\mathcal{H})}$ is consistent as $N \to \infty$, the probability of error between these two models is $0$.*

*Proof.* The first claim follows directly from Lemma B.2 and Dhir et al. (2024, Proposition 4.5). Given that $\mathcal{F}_\mathcal{G} \cap \mathcal{F}_\mathcal{H} = \emptyset$, only one of the models can be the true model. Consistency implies $\frac{P(X^N | \mathcal{M}_\mathcal{G})}{P(X^N | \mathcal{M}_\mathcal{H})} \to \infty$ if the data generating distribution $P \in \mathcal{F}_\mathcal{G}$, and $\frac{P(X^N | \mathcal{M}_\mathcal{G})}{P(X^N | \mathcal{M}_\mathcal{H})} \to 0$ if the data generating distribution $P \in \mathcal{F}_\mathcal{H}$ (Walker, 2004). Hence, the marginal likelihood is higher for the correct model for each distribution in the union of $\mathcal{F}_\mathcal{G} \cup \mathcal{F}_\mathcal{H}$ leading to a probability of error of $0$. $\square$

Consistency in the above requires certain conditions on the prior (Walker et al., 2004, Theorem 1). Suppose the data is generated from $X^N \sim P_0$, we require that for the correct model, suppose it is $\mathcal{M}_\mathcal{G}$, the prior puts non-negligible mass around $P_0$: $\pi_\mathcal{G}(\{P : \text{KL}[P_0 \| P] < \epsilon\}) > 0$ for all $\epsilon > 0$ (Walker et al., 2004). Second, the posterior for both models cannot paradoxically concentrate in a region of negligible prior mass: $\liminf_N \text{KL}[P_0 \| P_{NA}] \geq \epsilon$ for all $\epsilon > 0$ where $A := \{P : \text{KL}[P_0 \| P] > \epsilon\}$. $P_{NA}$ is the posterior (over $N$ samples) computed from the prior but restricted to the set of densities $A$, which are $\epsilon$ far from the true density $P_0$ as measured by the KL-divergence (Walker, 2004). This condition

stops the wrong model (which has negligible prior mass around the true density) from concentrating around the true density for a large enough sample size.

The condition for two causal models that $\mathcal{F}_\mathcal{G} \cap \mathcal{F}_\mathcal{H} = \emptyset$ also holds when the causal models are identifiable (Guyon et al., 2019, Ch. 2). Hence, Bayesian model selection can recover the causal structure if the causal models are identifiable.

**Corollary B.4.** *Assume two Bayesian causal models* $(\mathcal{M}_\mathcal{G}, \pi_\mathcal{G})$, *and* $(\mathcal{M}_\mathcal{H}, \pi_\mathcal{H})$ *where the underlying causal models are identifiable:* $\mathcal{F}_\mathcal{G} \cap \mathcal{F}_\mathcal{H} = \emptyset$. *If the estimate* $\frac{P(X^N|\mathcal{M}_\mathcal{G})}{P(X^N|\mathcal{M}_\mathcal{H})}$ *is consistent as* $N \to \infty$, *the probability of error between these two models is* 0.

Next, we consider the case of two Bayesian causal models where the two graphs are in the same MEC. Without restrictions, it is well known that in this case there may exist distributions in two Bayesian causal models that can express a given dataset (Pearl, 2009). Thus, the result from Lemma B.2 does not hold. The key idea that allows for distinction here is that the ICM assumption (as represented by factorised priors) may only hold in the causal factorisation of a Bayesian causal model. Further, the distributions of marginal likelihood values are sensitive to the ICM assumption. As the causal factorisation of the two Bayesian causal models differs, they will have a different distribution over marginal likelihood values.

We first formalise the ICM assumption as the separability of the priors with respect to a certain factorisation of the joint.

**Definition B.5.** (Dhir et al., 2024, Definition 4.2) Given a Bayesian causal model $(\mathcal{M}_\mathcal{G}, \pi_\mathcal{G})$, a prior $\pi$ is separable with respect to $\mathcal{C}_\mathcal{G}$ if it factorises $\prod_{i \in \mathcal{V}} \pi_i$ such that $\pi_i \in \mathcal{P}(\mathcal{C}_{i|\mathrm{PA}_\mathcal{G}(i)})$, where $\mathcal{P}$ is the set of all distributions over an object.

That is, if $\pi$ is separable with respect to $\mathcal{C}_\mathcal{G}$, the joint can be written as $\prod_{i \in \mathcal{V}} P_i(X_i|X_{\mathrm{PA}_\mathcal{G}(i)})\pi(\mathrm{d}P_i)$, with $P_i \in \mathcal{C}_{i|\mathrm{PA}_\mathcal{G}(i)}$. Separability of the prior (and hence the ICM condition) may hold in multiple factorisations. This is the case with normalised linear Gaussian models (Dhir et al., 2024, Appendix D.2). To formalise this, we reiterate the notion of *separability-compatibility*.

**Definition B.6.** (Dhir et al., 2024, Definition 4.6) Given two Bayesian causal models $(\mathcal{M}_\mathcal{G}, \pi_\mathcal{G})$, and $(\mathcal{M}_\mathcal{H}, \pi_\mathcal{H})$. Denote $\gamma : \mathcal{C}_\mathcal{G} \to \mathcal{C}_\mathcal{H}$ a bijection such that for any $P \in \mathcal{C}_\mathcal{G}$ such that $Q := \delta(P) \in \mathcal{C}_\mathcal{H}$, there holds an equality of joint measures $\prod_{i \in \mathcal{V}} P_i(X_i|X_{\mathrm{PA}_\mathcal{G}(i)}) = \prod_{i \in \mathcal{V}} Q_i(X_i|X_{\mathrm{PA}_\mathcal{H}(i)})$. The two Bayesian causal models are *separable-compatible* if: i) the pushforward $\pi_\mathcal{G} \circ \gamma^{-1}$ is separable with respect to $\mathcal{C}_\mathcal{H}$, ii) $\pi_\mathcal{H} \circ \gamma$ is separable with respect to $\mathcal{C}_\mathcal{G}$.

The operation $\delta$ is simply the transformation by Bayes' rule that transforms the factorisation according to $\mathcal{G}$ to the factorisation according to $\mathcal{H}$ while ensuring the joint is the same. Separability-compatibility checks whether the prior separates according to multiple factorisations. If it does, ICM holds in multiple factorisations.

**Theorem B.7.** *Given two Bayesian causal models* $(\mathcal{M}_\mathcal{G}, \pi_\mathcal{G})$, *and* $(\mathcal{M}_\mathcal{H}, \pi_\mathcal{H})$. *If* $\mathcal{G}$ *and* $\mathcal{H}$ *are in the same Markov equivalence classes (MEC), and the Bayesian causal models are not separable compatible, they cannot be Bayesian distribution-equivalent. Further, assuming data is generated from one of the models, then the probability of error is less than* 50%.

*Proof.* The first statement follows directly from Dhir et al. (2024, Proposition 4.7). The probability of error in the two model case can be written as (Dhir et al., 2024)

$$P(E) = \frac{1}{2}(1 - \mathrm{TV}[P(\cdot|\mathcal{M}_\mathcal{G}), P(\cdot|\mathcal{M}_\mathcal{H})]). \tag{17}$$

If the two models are not Bayesian distribution-equivalent, then $\mathrm{TV}[P(\cdot|\mathcal{M}_\mathcal{G}), P(\cdot|\mathcal{M}_\mathcal{H})] > 0$. This implies that the probability of error is less than $\frac{1}{2}$. $\qquad\square$

Theorem B.7 shows that if the ICM assumption only holds in the causal factorisation of the model (prior is only separable with respect to the causal factorisation), then there will always exist datasets that give different marginal likelihood values for the two Bayesian causal models. The above holds for any combination of Bayesian causal models with graphs in the same MEC. Hence, the total probability of error for the comparison within an MEC will be less than that of a graph randomly chosen from within the MEC (Friedman, 1996).

## C. CGP-CDE Details

For our Bayesian causal model, we use the Gaussian process conditional density estimator (Dutordoir et al., 2018). We use a sum of commonly used kernels Rasmussen (2003, Ch. 5), as outlined in Appendix C.1. We also apply variational inference

to optimise the lower bound, as is standard for latent variable Gaussian process models (Dutordoir et al., 2018; Titsias & Lawrence, 2010) and use inducing points with stochastic variational inference to improve data scalability (Hensman et al., 2013). We outline the variational inference in Appendix C.2 and detail our optimisation schedule in Appendix C.3.

### C.1. CGP-CDE Kernel

The ultimate aim of the approach is to find the causal graph that maximises the posterior probability. The choice of kernels will be problem dependent and it is possible to carry out Bayesian model selection over the kernels themselves (Rasmussen, 2003, Ch. 5). To allow for good fits of a range of datasets, we use a sum of kernels that can express functions with a range of lengthscales and roughness well.

The kernel for function $\mathbf{f}_i$ is

$$k_i = k_{\text{lin},i} + k_{\text{sqe},i} + k_{\text{m12},i} + k_{\text{m32},i} + k_{\text{rq},i}, \tag{18}$$

where each of the kernels is defined below (Williams & Rasmussen, 2006, Ch. 4). For all the kernels, $\theta_{ii}$ is the hyperparameter for the latent dimension.

$k_{\text{lin},i}$ is the linear kernel:

$$k_{\text{lin},i} = \sum_{j \neq i}^{D} \theta_{\text{lin},ij} x_j x'_j + \theta_{\text{lin},ii} w_i w'_i, \tag{19}$$

where $\theta_{\text{lin},ij}$ indicates the $j^{\text{th}}$ dimension of hyperparameter $\boldsymbol{\theta}_{\text{lin},i}$.

$k_{\text{sqe},i}$ is the squared exponential kernel

$$k_{\text{sqe},i} = \sigma_{\text{sqe},i}^2 \left( \sum_{j \neq i}^{D} \exp\left( -\theta_{\text{sqe},ij}^2 \frac{(x_j - x'_j)^2}{2} \right) + \exp\left( -\theta_{\text{sqe},ii}^2 \frac{(w_i - w'_i)^2}{2} \right) \right), \tag{20}$$

where $\theta_{\text{sqe}}$ is the precision parameter.

$k_{\text{m12},i}$ and $k_{\text{m32},i}$ are the Matérn12 and Matérn32 kernels with $\nu = \frac{1}{2}$ and $\nu = \frac{3}{2}$ respectively, with the general form

$$k_{\text{m}\nu,i} = \sigma_{\text{m}\nu,i}^2 \sum_{j \neq i}^{D} \frac{2^{1-\nu}}{\Gamma(\nu)} \left( \theta_{\text{m}\nu,ij} \sqrt{2\nu} |x_j - x'_j| \right)^\nu K_\nu \left( \theta_{\text{m}\nu,ij} \sqrt{2\nu} |x_j - x'_j| \right)$$
$$+ \sigma_{\text{m}\nu,i}^2 \frac{2^{1-\nu}}{\Gamma(\nu)} \left( \theta_{\text{m}\nu,ii} \sqrt{2\nu} |w_i - w'_i| \right)^\nu B_\nu \left( \theta_{\text{m}\nu,ii} \sqrt{2\nu} |w_i - w'_i| \right), \tag{21}$$

where $\Gamma(\nu)$ is the gamma functions and $B_\nu$ is a modified Bessel function. Similarly to the squared exponential kernel, $\theta_{\text{m}\nu}$ is the precision parameter.

$k_{\text{rq}}$ is the rational quadratic kernel, which is equivalent to the sum of many squared exponential kernels with different precision hyperparameters (Williams & Rasmussen, 2006, Ch. 4),

$$k_{\text{rq},i} = \sigma_{\text{rq},i}^2 \left( \sum_{j \neq i}^{D} \left( 1 + \theta_{\text{rq},ij}^2 \frac{(x_j - x'_j)^2}{2a_d} \right)^{-a_i} + \left( 1 + \theta_{\text{rq},ii}^2 \frac{(w_i - w'_i)^2}{2a_i} \right)^{-a_i} \right), \tag{22}$$

where $a$ is a hyperparameter that is learned.

The kernel variance terms $\sigma_{\text{sqe}}^2$, $\sigma_{\text{m}\nu}^2$ and $\sigma_{\text{rq}}^2$ mean individual kernels can be "switched off" by setting this term to zero if they do not contribute to improving the evidence lower bound.

In all these kernels, the hyperparameter denoted by $\theta$ controls the variability of the function, as discussed in subsection 4.2. This means as they tend to zero, the input and output of the function become decorrelated. To construct the adjacency matrix in Equation 10, we sum these hyperparameters, so that

$$\theta_{ij} = \theta_{\text{lin},ij} + \theta_{\text{sqe},ij} + \theta_{\text{m12},ij} + \theta_{\text{m32},ij} + \theta_{\text{rq},ij}. \tag{23}$$

## C.2. CGP-CDE Lower Bound

We use a variant on the GP-CDE (Dutordoir et al., 2018) to define our causal model and define a prior over distributions. Intractability in these models is commonly solved using variational inference (Lalchand et al., 2022; Dutordoir et al., 2018).

Each variable $\mathbf{x}_i$ is modelled as a function of all other variables $\mathbf{X}_{\neg i}$ and latent variable $\mathbf{w}_d$ plus some Gaussian noise, parameterised by noise variance $\phi_i^2$:

$$\mathbf{x}_i = \mathbf{f}_i(\mathbf{X}_{\neg i}, \mathbf{w}_i) + \varepsilon, \ \ \varepsilon \sim \mathcal{N}(0, \phi_i^2). \tag{24}$$

The presence of the latent variable allows learning of non-Gaussian and heteroscedastic noise (Dutordoir et al., 2018). A Gaussian process prior is placed on $\mathbf{f}_i$,

$$p(\mathbf{f}_i | \mathbf{X}_{\mathrm{PA}_{\mathcal{G}\mathbf{A}}(i)}, \mathbf{\Lambda}_i, \mathbf{w}_i) = \mathcal{N}(\mathbf{0}, K_{\mathbf{\Lambda}_i}((\mathbf{X}_{\neg i}, \mathbf{w}_i), (\mathbf{X}_{\neg i}, \mathbf{w}_i)')), \tag{25}$$

where $\mathbf{X}_{\mathrm{PA}_{\mathcal{G}\mathbf{A}}(i)}$ are the parents of $\mathbf{x}_i$ in $\mathbf{A}$, and $K$ is the kernel with set of hyperparameters denoted $\mathbf{\Lambda}_i$. We denote kernel hyperparameters that control dependence and are included in the adjacency matrix collectively as $\boldsymbol{\theta}$, while the rest as $\boldsymbol{\sigma}$, with $\Lambda = \{\boldsymbol{\theta}, \boldsymbol{\sigma}\}$.

Different prior beliefs of the form of $f$ can be expressed in the choice of kernel. To find the causal DAG with the highest posterior, we need to maximise the marginal likelihood of each variable. This is written as

$$\log p(\mathbf{x}_i | \mathbf{X}_{\mathrm{PA}_{\mathcal{G}\mathbf{A}}(i)}, \mathbf{\Lambda}_i, \boldsymbol{\phi}_i) = \log \int \int p(\mathbf{x}_i | \mathbf{f}_i, \boldsymbol{\phi}_i) p(\mathbf{f}_i | \mathbf{X}_{\mathrm{PA}_{\mathcal{G}\mathbf{A}}(i)}, \mathbf{\Lambda}_i, \mathbf{w}_i) p(\mathbf{w}_i) d\mathbf{w}_i d\mathbf{f}_i.$$

However, the latent variables $\mathbf{w}_i$ appear non-linearly in $p(\mathbf{f}_i | \mathbf{X}_{\mathrm{PA}_{\mathcal{G}\mathbf{A}}(i)}, \mathbf{\Lambda}_i, \mathbf{w}_i)$ making the calculation of $\log p(\mathbf{x}_i | \mathbf{X}_{\mathrm{PA}_{\mathcal{G}\mathbf{A}}(i)}, \mathbf{\Lambda}_i, \boldsymbol{\phi}_i)$ analytically intractable.

The addition of inducing points $\mathbf{u}_i$, with corresponding inducing inputs $\boldsymbol{Z}_i$, helps both with the intractability of $\log p(\mathbf{x}_i | \mathbf{X}_{\mathrm{PA}_{\mathcal{G}\mathbf{A}}(i)}, \mathbf{\Lambda}_i, \boldsymbol{\phi}_i)$ and scaling to large datasets (Titsias, 2009; Titsias & Lawrence, 2010). We can write the joint between $\mathbf{f}$ and $\mathbf{u}_i$ as:

$$p\left(\begin{bmatrix} \mathbf{f}_i \\ \mathbf{u}_i \end{bmatrix}\right) = \mathcal{N}\left(\begin{bmatrix} \mathbf{0} \\ \mathbf{0} \end{bmatrix}, \begin{bmatrix} K_{i,\mathbf{ff}} & K_{i,\mathbf{fu}} \\ K_{i,\mathbf{uf}} & K_{i,\mathbf{uu}} \end{bmatrix}\right), \tag{26}$$

where $K_{\mathbf{ff}}$ is the covariance matrix between training data, $K_{\mathbf{uu}}$ is the covariance matrix between inducing points and $K_{\mathbf{fu}}$ and $K_{\mathbf{uf}}$ are the covariance matrices between the two. Suppressing the hyperparameters and $\boldsymbol{Z}_i$ for brevity, the marginal likelihood can then be written as:

$$\log p(\mathbf{x}_i | \mathbf{X}_{\mathrm{PA}_{\mathcal{G}\mathbf{A}}(i)}) = \log \int \int p(\mathbf{x}_i | \mathbf{f}_i) p(\mathbf{f}_i | \mathbf{X}_{\mathrm{PA}_{\mathcal{G}\mathbf{A}}(i)}, \mathbf{w}_i, \mathbf{u}_i) p(\mathbf{w}_i) p(\mathbf{u}_i) d\mathbf{w}_i d\mathbf{f}_i d\mathbf{u}_i. \tag{27}$$

where $p(\mathbf{u}_i) = \mathcal{N}(\mathbf{0}, K_{i,\mathbf{uu}})$. This is still not tractable, so we use variational inference to define an evidence lower bound (ELBO) to the marginal likelihood (Titsias, 2009). This is done by introducing a variational distribution $q(\mathbf{f}_i, \mathbf{w}_i, \mathbf{u}_i)$ and rewriting the marginal likelihood as (suppressing conditioning terms for neatness):

$$\log p(\mathbf{x}_i) = \log \int \int \int p(\mathbf{x}_i | \mathbf{f}_i, \boldsymbol{\phi}_i) p(\mathbf{f}_i | \mathbf{u}_i, \mathbf{X}_{\mathrm{PA}_{\mathcal{G}\mathbf{A}}(i)}, \mathbf{w}_i) p(\mathbf{u}_i) p(\mathbf{w}_i) \frac{q(\mathbf{f}_i, \mathbf{w}_i, \mathbf{u}_i)}{q(\mathbf{f}_i, \mathbf{w}_i, \mathbf{u}_i)} d\mathbf{u}_i d\mathbf{w}_i d\mathbf{f}_i,$$

The variational distribution takes the form (Titsias & Lawrence, 2010; Dutordoir et al., 2018):

$$q(\mathbf{f}_i, \mathbf{w}_i, \mathbf{u}_i) = p(\mathbf{f}_i | \mathbf{w}_i, \mathbf{u}_i) q(\mathbf{w}_i) q(\mathbf{u}_i), \tag{28}$$

where $q(\mathbf{u}_i) = \mathcal{N}(\mathbf{u}_i | \boldsymbol{m}_{u,i}, \boldsymbol{S}_{u,i})$ is the Gaussian variational distribution of inducing points $\mathbf{u}_i$ and $q(\mathbf{w}_i) = \mathcal{N}(\boldsymbol{\mu}_i, \boldsymbol{\Sigma}_i)$ is the variational distribution of $\mathbf{w}_i$.

Rearranging and using Jensen's inequality, we can then get the lower bound for variable $d$ as (Dutordoir et al., 2018)

$$\log p(\mathbf{x}_i) \geq \left\langle \langle \log p(\mathbf{x}_i | \mathbf{f}_i) \rangle_{q(\mathbf{f}_i)} \right\rangle_{q(\mathbf{w}_i)} - KL[q(\mathbf{u}) || p(\mathbf{u})] - KL[q(\mathbf{w}_i) || p(\mathbf{w}_i)], \tag{29}$$

where $q(\mathbf{f}_i) = \int p(\mathbf{f}_i | \mathbf{u}, \mathbf{w}_i) q(\mathbf{u}) d\mathbf{u}$. We denote the lower bound for variable $X_i$ as $\mathcal{L}_{\text{ELBO},i}(q_i, \boldsymbol{\Lambda}_i, \phi_i)$. The full lower bound then is $\mathcal{L}_{\text{ELBO}}(q, \boldsymbol{\Lambda}_i, \phi_i) := \sum_{i=1}^{D} \mathcal{L}_{\text{ELBO},i}(q_i, \boldsymbol{\Lambda}_i, \phi_i)$, remembering $\boldsymbol{\Lambda} = \{\boldsymbol{\theta}_i, \boldsymbol{\sigma}_i\}$.

By incorporating the optimal Gaussian variational distribution $q(\mathbf{u})$, which can be calculated in closed form, it is possible to collapse the bound by integrating out the inducing variables (Titsias, 2009; Titsias & Lawrence, 2010). However, the computational complexity of this approach is $\mathcal{O}(NM^2D)$, meaning for large values of $N$ training becomes infeasible (Lalchand et al., 2022). To allow our method to scale to large datasets, we follow the approach of Hensman et al. (2013) and keep $q(\mathbf{u})$ uncollapsed so that stochastic variational inference can be used. This means the data can be mini-batched in training, reducing the computational complexity to $\mathcal{O}(M^3D)$. To further improve computational efficiency, we use an encoder to learn a function $g_{q,i} : (\mathbf{x}_{\text{batch}}) \mapsto (\mu_i, \Sigma_i)$ for the variational distribution $q(\mathbf{w}_i) = \mathcal{N}(\mu_i, \Sigma_i)$, where $\mathbf{x}_{\text{batch}} \in \mathbb{R}^{b \times D}$ is a batch of data points, $\mu_i \in \mathbb{R}^b$ and $\Sigma_i \in \mathbb{R}^b$ for batch size $b$ (Dutordoir et al., 2018). This is more efficient than the alternative of learning $N$ parameters. We use a multi-layer perceptron for the encoder and list the hyperparameters used in Table 1.

Table 1: Hyperparameters for the variational encoder $g_{q,i} : (\mathbf{x}_{\text{batch}}) \mapsto (\mu_i, \Sigma_i)$.

| Parameter | Value |
|---|---|
| hidden layer size | 128 |
| number of layers | 5 |
| activation function | ReLU |

**Other hyperparameters:**   We follow common practice and maximise the lower bound with respect to the hyperparameters $\boldsymbol{\sigma}$ and $\phi$ as well (Rasmussen, 2003) (along with $\boldsymbol{\theta}$ and $q$). This is justified based on the fact that the posterior for these hyperparameters tends to be highly peaked in practice (MacKay, 1999). Thus, a Laplace approximation can be approximated by the maximum value.

### C.3. Optimisation schedule

We split the training schedule up into three steps. The warm-up phase allows the model to learn conditional independences present in the data. In the acyclic constraint phase, where a penalty method is used to enforce the DAG constraint, the model then learns the direction of any edges. The learnt DAG is then enforced, and the ELBO is optimised in the cool-down phase to find the ELBO of the final model, which can then be used for comparing random restarts.

**Warm-up phase:**   The warm-up phase allows the model to learn any conditional independences from the data. It also ensures the hyperparameters and variational parameters of the Gaussian process are at reasonable values at the beginning of the acyclic phase. Although the number of iterations needed will depend on the problem at hand, we found $T_0 = 25,000$ iterations for this phase sufficient to ensure convergence.

**Acyclic constraint phase:**   After the warm-up phase has concluded, a lot of the edges corresponding to conditional independences are switched off. However, for variables that are correlated, the causal direction has not been decided. This part of the training forces the constraint that the final graph has to be acyclic. The loss is

$$\mathcal{L}_{\text{ELBO}}(q, \boldsymbol{\theta}, \boldsymbol{\sigma}, \phi) := \mathcal{L}_{\text{ELBO}}(q, \boldsymbol{\theta}, \boldsymbol{\sigma}, \phi) + \log p(\boldsymbol{\theta}) - \gamma_t h(\mathbf{A}), \tag{30}$$

is optimised, using the acyclic regulariser proposed by Nazaret et al. (2024). The weighting $\gamma_t$ of the acyclic penalty term $h(\mathbf{A})$ is increased linearly each epoch by $\rho = 50$, starting from 0. As we use minibatching, one epoch is a full pass through the dataset. For some of the 50 variable datasets, we increased this to $\rho = 250$ to reduce the time required to train. This is because the unconstrained loss scales with the number of variables, so for the 50 variable dataset, a higher $\rho$ ensures a higher ratio between the acyclic constraint and the rest of the loss, ensuring faster convergence.

We use 50 power iterations to approximate the eigenvectors required to calculate the gradients of the acyclic constraint, as described in Appendix A.2. The acyclic constraint phase is terminated when $h(\mathbf{A}_{\boldsymbol{\theta}_t}) < \tau$ where $\tau = 0.005$, or the number of iterations exceeds a maximum value of $T_a = 50,000$.

**Acyclic thresholding:** After the end of the acyclic phase, the values of the hyperparameters are sometimes not exactly 0 due to numerical precision, we find the adjacency matrix with an extra thresholding step. Following Lachapelle et al. (2019), we threshold weights to 0, starting with the lowest weight, until the adjacency is exactly acyclic ($h(\mathbf{A}_{\boldsymbol{\theta}_t}) = 0$).

**Cool-down phase:** We found that random restarts helped tackle local optima in optimisation Appendix C.4. To allow for the comparison of ELBO values across restarts (see Appendix C.4) we continue training the found acyclic DAG for $T_f$ iterations. This ensured that the found ELBO approximates the ELBO of the final resultant DAG. As some hyperparameters may have been pushed to extreme values by the acyclic threshold if competing with an anti-causal edge, we reinitialise the hyperparameters of the edges that are still active to their initial value. This reduces the number of iterations needed for the cool-down phase. We found that $T_f = 25,000$ cool-down iterations sufficed here.

**Final matrix thresholding:** Finally, we perform an extra thresholding step to remove edges that do not encode correlations. As our data is normalised, we check the kernel evaluation of points $-3$ and $3$, which should bound $95\%$ of the data. If the function is highly correlated between these points, it is effectively constant for $95\%$ of the data. We thus threshold the linear variances to $10^{-4}$ and the graph parameters (the rest of the kernels) to $0.05$, which corresponds to a lengthscale of $20$. These values ensure that points between $-3, 3$ are highly correlated.

### C.4. Random Restarts

The main principle behind our method is to find the graph that maximises the marginal likelihood (or a lower bound to it). Maximising the ELBO with respect to the hyperparameters $\mathbf{\Lambda}$ in Gaussian process models is known to suffer from local optima issues (Rasmussen, 2003, Ch. 5). The final hyperparameters found, and hence the adjacency matrix, can be dependent on the initialisation of the hyperparameters. To mitigate this, we use a widely adopted technique of performing random restarts (Dhir et al., 2024). We optimise the loss $\mathcal{L}$ starting from multiple initialisations (this can be done in parallel) $N_r$ times. Then we pick the graph that achieves the highest $\mathcal{L}_{\mathrm{ELBO}}$ as the candidate for the most likely graph. Adding more random restarts for the CGP-CDE should improve performance, as shown for the DGP-CDE Appendix D.1.

---

**Algorithm 1** Optimisation procedure for the Causal GP-CDE.

---

**Input:** Data $\mathbf{X}$, number of random restarts $N_r$, acyclic penalty weighting update $\rho$, threshold $\tau$, acyclic penalty weighting $\gamma_t = 0$ initial $\mathbf{\Lambda} = \{\boldsymbol{\theta}, \boldsymbol{\sigma}\}$

**Result:** Most likely adjacency matrix $\mathbf{A}^*$

Initialise empty list $graphscore$

**for** $i = 1$ **to** $N_r$ **do**
    **for** $j = 1$ **to** $T_0$ **do**
        Update $q, \mathbf{\Lambda}$ by maximising $\mathcal{L}$ (Equation (30))
    **end**
    $t \leftarrow 0$
    **while** $h(\mathbf{A}_{\boldsymbol{\theta}_t}) > \tau$ and $t < T_a$ **do**
        Update $q, \mathbf{\Lambda}$ by maximising $\mathcal{L}$ (Equation (30))
        $t \leftarrow t + 1$
        **if** *epoch ended* **then**
            $\gamma_t \leftarrow \gamma_t + \rho$
        **end**
    **end**
**end**
Set $\gamma_t$ to $0$
Threshold $\mathbf{A}_{\boldsymbol{\theta}_t}$ starting from lowest weight until $h(\mathbf{A}_{\boldsymbol{\theta}_t}) = 0$
**for** $k = 1$ **to** $T_f$ **do**
    Update $q, \mathbf{\Lambda}$ by maximising $\mathcal{L}$ (Equation (30))
**end**
Append $(\mathbf{A}_{\boldsymbol{\theta}}, \mathcal{L})$ to $graphscore$.
$\mathbf{A}^* \leftarrow \mathbf{A}_{\boldsymbol{\theta}}$ from $graphscore$ with the maximum $\mathcal{L}$.

---

## C.5. Hyperparameter Priors

As discussed in Section 4.3, we can place priors on the graphs by placing priors on the hyperparameters $\boldsymbol{\theta}$ of the CGP-CDE. For our implementation, we place a prior on $\boldsymbol{\theta}$ that favours sparser graphs. Specifically, we use the Gamma prior $P(\boldsymbol{\theta}) = \mathrm{Gamma}(\eta, \beta)$, where $\eta$ is the shape parameter and $\beta$ is the rate parameter. For all experiments, we set $\eta = 1$ and $\beta = 10$. Our hyperparameters for the priors were heuristically chosen and give log probabilities that are a fraction of the rest of the loss.

## C.6. Implementation Details

In this section, we outline the details of our implementation.

**Hyperparameter initialisations**  We initialise the hyperparameters $\boldsymbol{\theta} \sim \mathrm{Uniform}(0.01, 1)$, except $\theta_{\mathrm{lin}} = 0.25$. All kernel variances $\boldsymbol{\sigma}$ are initialised to 1 so they all have the same initial weighting and likelihood variance initialised as $\phi^2 = \frac{1}{\kappa^2}$ where $\kappa \sim \mathrm{Uniform}(50, 100)$. The $a$ term in the rational quadratic kernel is initialised as $a \sim \mathrm{Uniform}(0.1, 10)$.

**Variational parameters**  We use 400 inducing points for all datasets as we found this a reasonable trade off between computation time and accuracy. We initialise our inducing point locations at a subset of the data inputs. We use a multilayer perception for the latent variable encoder, as described in Appendix C.2, with hyperparameters listed in Table 1. The encoder weights are initialised from a truncated normal distribution centred on 0 with a standard deviation of $\sqrt{\left(\frac{2}{\text{hidden layer size}}\right)}$ (He et al., 2015).

**Loss calculation**  We use a minibatch size of 256 to calculate the loss. We take 50 Monte Carlo samples to integrate out the latent variables $\mathbf{w}_i = \mathcal{N}(\mu_i, \Sigma_i)$.

**Acyclic constraint penalty**  The acyclic constraint penalty has three hyperparameters: the coefficient to linearly increase the weighting of the penalty term by, $\rho$, the threshold value for the acyclicity, $\tau$, and the number of power iterations used to approximate the eigenvectors for the gradients of the acyclic constraint. We tuned these variables using five datasets generated from 10 variable ER1.5 graphs. We chose 50 power iterations, as it led to better performance than smaller numbers of iterations and after 50 we saw minimal improvement. For the 10 variable ER1.5 datasets, we found $\rho = 50$ and $\tau = 0.005$ leads to the best performance across SHD, SID and F1, while still converging in a reasonable amount of time. However, as the number of variables scaled, this value of $\rho$ took prohibitively long, except for the Syntren dataset. This is because the ELBO scales with the number of variables, $\mathcal{L}_{\mathrm{ELBO}}(q, \boldsymbol{\Lambda}_i, \phi_i) := \sum_{i=1}^{D} \mathcal{L}_{\mathrm{ELBO},i}(q_i, \boldsymbol{\Lambda}_i, \phi_i)$. Therefore, for the 3, 20 and 50 variable datasets we scale $\rho$ with the number of variables, such that $\rho = 5D$. This ensures the relative magnitude of the ELBO and the acyclic penalty remains roughly the same, meaning the time for the acyclic constraint phase to converge is not dependent on the number of variables. We did not find scaling $\rho$ affected the performance, but it did speed up the acyclic phase.

**Optimisers**  Each optimisation step consists of two steps. First, we optimise the parameters of the variational distribution of inducing points $q(\mathbf{u}_i) = \mathcal{N}(\mathbf{u}_i | \boldsymbol{m}_{u,i}, \boldsymbol{S}_{u,i})$ using natural gradients, which have been shown to significantly improve training time for variational Gaussian processes (Salimbeni et al., 2018). We linearly increase the natural gradient step size from 0.0001 to 0.1 for the first five iterations, and then use a step size of 0.1. Second, we optimise the Gaussian process hyperparameters using Adam with a learning rate of 0.05 (Kingma & Ba, 2014).

**Random restarts**  We did as many random restarts as we had computational resources for. This means we did 2 restarts for the three-variable dataset, 2 restarts for each of the 50 variable datasets and the Syntren dataset, and 1 restart for the 20 variable datasets.

# D. DGP-CDE Details

In Section 6.1 we introduce the discrete Gaussian process conditional density estimator (DGP-CDE), which is used to enumerate every possible causal structure for the 3 variable case. For this, a separate DGP-CDE is trained for each of the 25 possible causal graphs, with the Gaussian process prior for each variable defined in Equation (9). The DGP-CDE differs from the CGP-CDE in that it only takes $\mathbf{X}_{\mathrm{PA}_{\mathcal{G}}(i)}$ as inputs, whereas the CGP-CDE takes $\mathbf{X}_{\neg i}$ as inputs and then learns the

adjacency matrix.

As the adjacency matrix is fixed for the DGP-CDE there is no acyclic penalty, so the loss function is just the lower bound $\mathcal{L}_{\text{ELBO}}(q, \boldsymbol{\Lambda}, \boldsymbol{\phi}) := \sum_{i=1}^{D} \mathcal{L}_{\text{ELBO},i}(q_i, \boldsymbol{\Lambda}_i, \phi_i)$. When all 25 separate DGP-CDE models have been trained, the most likely graph is selected by choosing the model with the highest marginal likelihood.

### D.1. Implementation Details

**Loss Calculation** As the DGP-CDE is limited in the number of variables it can model by the need to enumerate over all graphs, there is no need to make the lower bound scalable using the methods discussed in Appendix C.2 (Hensman et al., 2013). Instead, we use the collapsed version of the lower bound, which can be calculated in closed form (Titsias, 2009).

**Kernels** We use a sum of a linear kernel defined in Equation (19) and a squared exponential kernel defined in Equation (20). This kernel means the kernel expectations needed to calculate the lower bound can be computed in closed form.

**Hyperparameters** The kernel variances are initialised as $\boldsymbol{\sigma}_i = 1$, the likelihood variance is randomly sampled as $\phi_i^2 \sim \text{Uniform}(10^{-4}, 10^{-2})$ and the precision parameter $\boldsymbol{\theta}_i \sim \text{Uniform}(1, 100)$.

**Variational parameters** As the enumeration over graphs is computationally expensive, we use 200 inducing points, which we found to be sufficient. As we are not using stochastic variational inference (Hensman et al., 2013) for the DGP-CDE, we can no longer mini-batch, so don't use a multilayer perceptron for the latent variables. Instead, we initialise the latent variable mean as $\mu_i = 0.1\mathbf{x}_i$ and standard deviation is randomly sampled $\Sigma_i \sim \text{Uniform}(0, 0.1)$.

**Optimisation Schedule** The use of the collapsed bound for the DGP-CDE requires a different optimisation schedule than for the CGP-CDE. This consists of a two part optimisation scheme. Due to the loss function being highly non-convex and suffering from local optima, the first step of the optimisation scheme uses Adam (Kingma & Ba, 2014) with a learning rate of 0.05 to get into a good region of the decision space. Once the loss function reaches the value a noise model would have, the optimisation scheme switches to the Broyden-Fletcher-Goldfarb-Shanno (BFGS) algorithm to perform gradient descent for the final part of the optimisation.

**Random Restarts** We did 10 random restarts for each DGP-CDE, selecting the one with the best lower bound for our final result. We analysed the effect of the random restarts by permuting the order of the random restarts and plotting, in Figure 3, the mean and standard deviation of the SHD, SID and F1 across permutations as the number of restarts increases. From the plot, it is clear that as the number of random restarts increases, the mean value of the metrics, on average, improves. This is because more random restarts allow a more thorough exploration of the loss function. It also shows that a higher value of $\mathcal{L}_{\text{ELBO}}$ does lead to better causal structure discovery.

## E. Data Standardisation

For additive noise models, it has been shown that when data is sampled from a simulated DAG and not normalised, the marginal variance tends to increase along the causal order (Reisach et al., 2021). When this is the case, it is possible to, at least partially, determine the causal order by ranking the marginal variance of variables. Reisach et al. (Reisach et al., 2021) demonstrate that commonly used synthetic datasets often have this property, meaning a simple baseline method based on variance sorting and regression can perform as well as some recent continuous structure learning algorithms. This variance can be manipulated by rescaling (such as changing measurement units), which would then give a different causal order. As such, results that take advantage of the variance may be sensitive to measurement scales. Such effects may also persist in real-world datasets due to the measurement scale used.

To get rid of these effects, we standardise all datasets before applying any of the causal discovery methods discussed in this paper. For the synthetic datasets, we also normalise each variable during the data generating process. This has been shown to create graphs that are not Var-sortable or $R^2$-sortable, meaning the causal order cannot be determined using variances and correlation artefacts (Ormaniec et al., 2024). This means our results for the baselines don't match the results reported in the original papers for similar datasets (Montagna et al., 2023; Lachapelle et al., 2019; Rolland et al., 2022) (note that some of these works also define SHD of an anti-causal edge as 1 where we define it as 2).

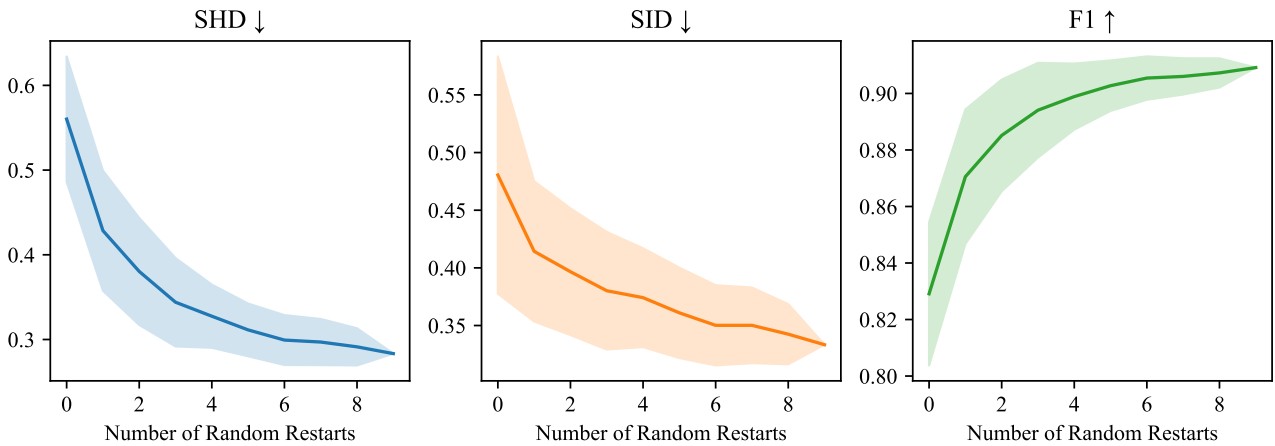

Figure 3: Effect of adding random restarts for the DGP-CDE. We record the metrics (SHD, SID and F1) for the restart with the lowest loss out of a growing set of restarts, starting with one restart and adding one restart at a time. We randomly permute the order in which the random restarts are added 50 times and plot the mean and standard deviation for each of the metrics. This shows that, on average, increasing the number of random restarts improves performance.

## F. Data Generation Details

**3 Variable Synthetic Data:** For each of the six distinct causal structures for three variables (up to permutations of the variables), we generated five datasets, each with 1000 samples. The data was generated by sampling a GP-CDE with a sum of a linear kernel and one other kernel randomly selected out of Matérn12, Matérn32, Matérn52, squared exponential and rational quadratic. The kernel hyperparameters are sampled from $\sigma_i \sim \text{Uniform}(1, 100)$ and $\theta_i \sim \text{Gamma}(1.5, 1)$ while the other parameters are sampled as $\phi_i \sim \text{Uniform}(0.01, 1)$ and $\mathbf{w}_i \sim \mathcal{N}(0, 1)$. The data is then sampled from this GP-CDE.

**20 & 50 Variable Synthetic Data:** For both 20 and 50 nodes, we create synthetic graphs using the ER (Erdos et al., 1960) and SF (Barabási & Albert, 1999) sampling schemes, with expected edges of one and four to simulate sparse and dense graphs. For each graph type, we generate five random graphs. Data is then generated for each node $X_i$ using

$$X_i := f_i^{NN}(\mathbf{X}_{\text{PA}_{\mathcal{G}}(i)}, \epsilon_i), \tag{31}$$

where $\text{PA}_{\mathcal{G}}(i)$ are the parents of $X_i$ and $\epsilon_i \sim \mathcal{N}(0, 1)$ is sampled from $\epsilon_i \sim \mathcal{N}(0, 1)$. $f^{NN}$ is a randomly initialised neural network with two layers, 128 units and ReLU activation functions. For each graph, we sample 1000 data points. This data generating scheme ensures that the final data is generated from a complicated distribution which does not directly correspond to any of the models.

**Syntren Dataset:** Syntren is a synthetic data generator that approximates real transcriptional regulatory networks (Van den Bulcke et al., 2006). Networks are selected from previously described transcriptional regulatory networks. Relationships between the genes are based on the Michaelis-Menten and Hill enzyme kinetic equations. The kinetic equations contain some biological noise and some lognormal noise is added on top. We use the data generated by Lachapelle et al. (2019). This contains 10 datasets of 500 samples, each with 20 nodes. The data was generated using the syntren data generator (Van den Bulcke et al., 2006), using the E. coli network with the default parameters except for the *probability for complex 2-regulator interactions*, which was set to one (Lachapelle et al., 2019).

## G. Details about Baselines

**SCORE & NoGAM:** For SCORE (Rolland et al., 2022) and NoGAM (Montagna et al., 2023) we use the implementation in the dodiscover package https://github.com/py-why/dodiscover. SCORE and NoGAM have very similar hyperparameters, and for both methods, we use the values of ridge regression suggested by Montagna et al., which they

tuned to minimise the generalisation error on the estimated residuals (Montagna et al., 2023). We also use the values suggested by the authors for both methods for all the hyperparameters for the Stein gradient, Hessian estimators and CAM pruning step except for the cutoff value for the CAM pruning step which we set to 0.01 as is standard for CAM and follows Montagna et al. (Montagna et al., 2023). NoGAM has the additional hyperparameter of *number of cross validation models*, for which we use the preset value of five.

**CAM:** For CAM, we preliminary neighbourhood selection and DAG pruning method with a cutoff value of 0.001 for the edge pruning as suggested by the authors (Bühlmann et al., 2014). We used the implementation in the `https://github.com/kurowasan/GraN-DAG` repository.

**NOTEARS:** For NOTEARS, we used the implementation provided in `https://github.com/azizilab/sdcd`, which uses binary search to find the adjacency matrix threshold that gives the largest possible DAG. This procedure has been found to give better performance than the default threshold of 0.3 in the original NOTEARS paper (Nazaret et al., 2024; Lopez et al., 2022).

**SDCD:** Nazaret et al. did not tune their hyperparameters for every dataset, instead using the same hyperparameter values, which were found to perform well empirically, for all their experiments (Nazaret et al., 2024, Appendix C.2). We use these same hyperparameter values and their implementation provided in `https://github.com/azizilab/sdcd`.

**DiBS:** We use the non-linear model provided by the authors in `https://github.com/larslorch/dibs/`. For graph priors, we use the ER prior for the ER graphs and SF prior for the SF graphs, Syntren and the three variable baseline. This is because SF more closely matches the structure of the Syntren and the three variable datasets. For the other hyperparameters we use the values fine-tuned by the authors, including an observational noise of 0.1 and 3000 optimisation steps. The authors fine-tuned different values for the bandwidths of the kernel and slope of the linear schedule depending on the number of variables. We use these values, which are listed in (Lorch et al., 2021, Appendix E.3). DiBS typically returns a posterior over DAGs using Stein variational gradient descent. When a single particle is used, this becomes the MAP estimate of the graph. Therefore, as we wish to compare to our method, which returns a MAP estimate, we use one particle for DiBS, and perform five random restarts, selecting the one with the highest marginal likelihood value.

**Random:** For the three variable dataset, we include uniform random sampling of DAGs. To compute this, for each ground truth graph, we enumerate all 25 possible graphs and evaluate their performance metrics. This exhaustive enumeration ensures that our results approximate those obtained in the limiting case where the number of sampled DAGs tends to infinity.

## H. Code Availability & Computational Resources

Code and data to replicate the experiments in this paper can be found at `https://github.com/Anish144/ContinuousBMSStructureLearning`. The experiments in this paper were run on A100 and RTX 4090 GPUs.

## I. Additional Experiments

### I.1. 20 Variable ER4 Comparison with Additive Noise Model

Our model is more expressive than previous methods, many of which rely on the additive noise assumption. Here, we answer the following question: If the dataset is generated from an ANM model, how does our more expressive model perform?

To test this, we introduce the CGP, which is the same as the CGP-CDE model (Section 4), except that it is restricted to an additive noise model. This is done by simply removing the latent variable $\mathbf{w}$ in Equation (8). The likelihood then is a Gaussian, hence an ANM model. Removal of the latent variable also means that inference is more accurate in this model than the CGP-CDE, however at the cost of a loss of flexibility. All the settings of the CGP model are exactly the same as in Appendix C.

Figure 4a shows the performance on five datasets generated from 20 variable ER4 graphs with Gaussian process functional relations. Thus, the datasets were generated from an additive noise model. It is worth noting that the CGP outperforms other methods that make the ANM assumption, such as CAM, SCORE, NOGAM, DiBS, and SDCD. Despite being more expressive than the CGP model, the CGP-CDE model performs on par with the CGP model. This shows that with the

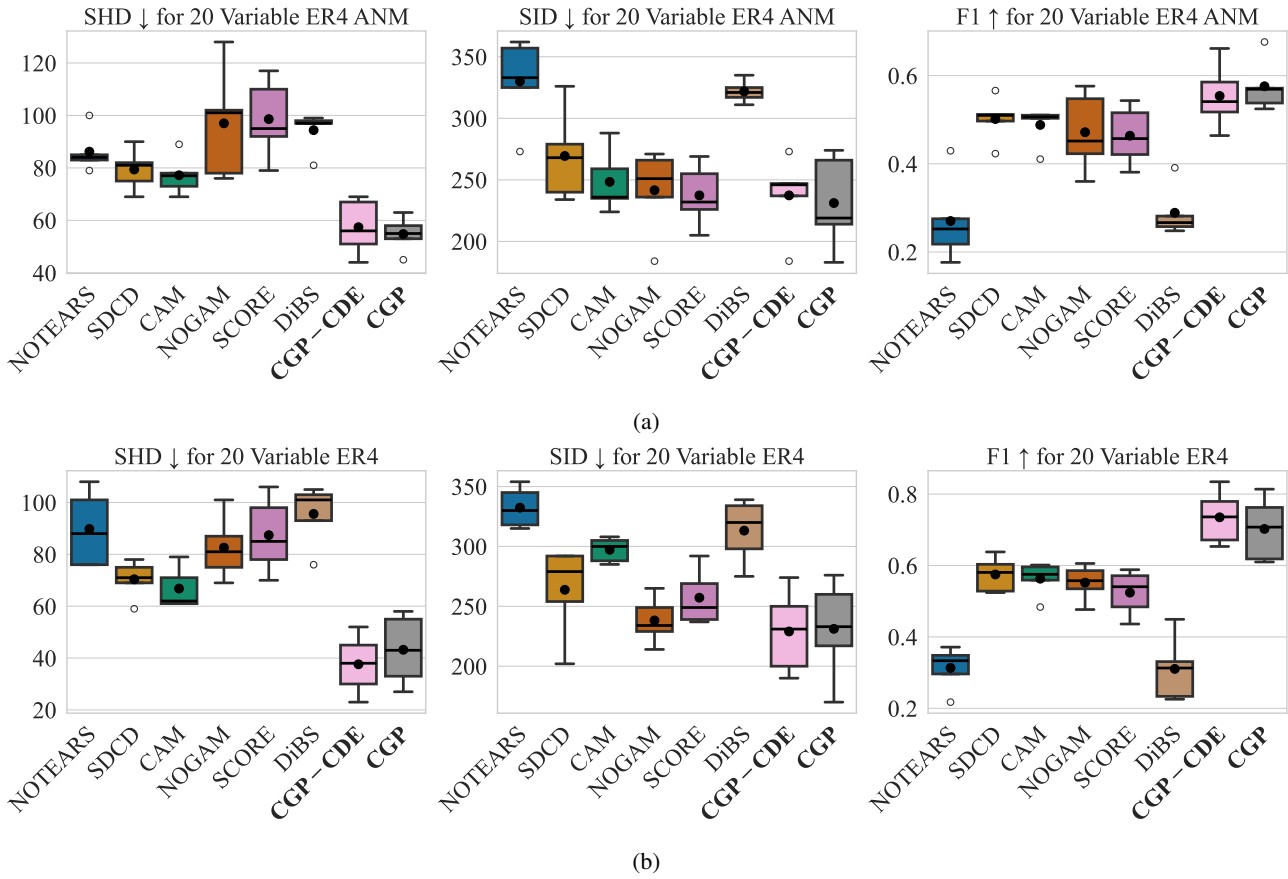

Figure 4: 20 variable ER4 graphs, with data generated from (a) an additive noise model (Gaussian process) and (b) a non-additive noise model (neural network) as described by Equation (31). Black dot is the average metric. Lower is better for SHD and SID, and higher is better for F1. Despite the CGP-CDE being more expressive than the CGP, it is still able to perform comparably for the ANM dataset, demonstrating the added flexibility of the CGP-CDE does not hinder the performance on datasets generated from restricted models.

added flexibility of our model, we do not lose performance on datasets generated from restricted models. This is due to the Occam's razor effect of Bayesian model selection, which allows for expressing simpler models when the data requires it (Rasmussen & Ghahramani, 2000).

For comparison, Figure 4b shows results for five 20 variable ER4 non-additive noise datasets generated using a neural network as described in Equation (31). Here, the CGP-CDE outperforms the CGP due to its more flexible functional assumptions. Also, the CGP outperforms the other ANM methods, showing the benefit of the Bayesian model selection approach even when restrictive functional assumptions are made.

### I.2. 20 Variable with Different Numbers of Samples Results

To investigate the effect of the number of samples on the performance of the models, we performed experiments on five 20 variable ER4 graphs, with 100, 500 and 1000 random samples. These results are shown in Figure 5. The CGP-CDE generally outperforms the other methods for metrics and numbers of samples, except for with 100 samples, where CAM performs slightly better and NoGAM and SCORE perform comparatively, however, the CGP-CDE outperforms these methods on SID and F1.

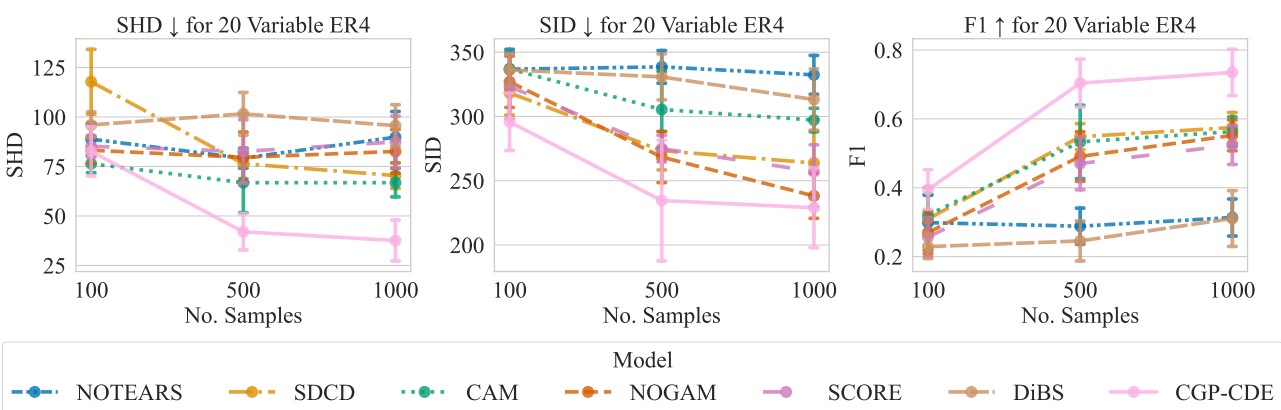

Figure 5: 20 variable ER4 graphs with data generated for 5 graphs as described by Equation (31). The x axis is the number of samples per graph, the circular markers are the mean and the vertical lines show the standard error across graphs. Lower is better for SHD and SID, higher is better for F1.

### I.3. 20 Variable Results

To show the CGP-CDE performs well with a smaller number of variables on a range of graph types and densities, we performed experiments on 20 variable graphs with the following graph types and densities: ER1 (Figure 6a), ER4 (Figure 4b), SF1 (Figure 6b), and SF4 (Figure 6c). These results show the CGP-CDE consistently performs well, although it is outperformed on some metrics for the ER1 and SF1 graphs, particularly by NOGAM, which seems to perform better on sparser graphs, perhaps because it is an ordering based method. However, the CGP-CDE is more consistent in performance than NOGAM across all our experiments with varying number of variables and graph types.

### I.4. Full 50 Variable Results

To demonstrate the performance of the CGP-CDE on a range of graph types and densities, we performed experiments on 50 variable graphs with the following graph types and densities: ER1, ER4, SF1, and SF4. The results of these experiments can be seen in Figure 7. These results show the CGP-CDE consistently performs well. For the ER graphs, the CGP-CDE outperforms all the baselines on SHD, SID and F1. For the SF graphs, where a few variables cause the rest of the variables, some of the baselines outperform the CGP-CDE on SID. We hypothesise this is because the structure of the SF graphs benefits methods that use topological search, such as NOGAM and SCORE. However, the CGP-CDE greatly outperforms these methods on SHD and F1 for both the SF graph densities.

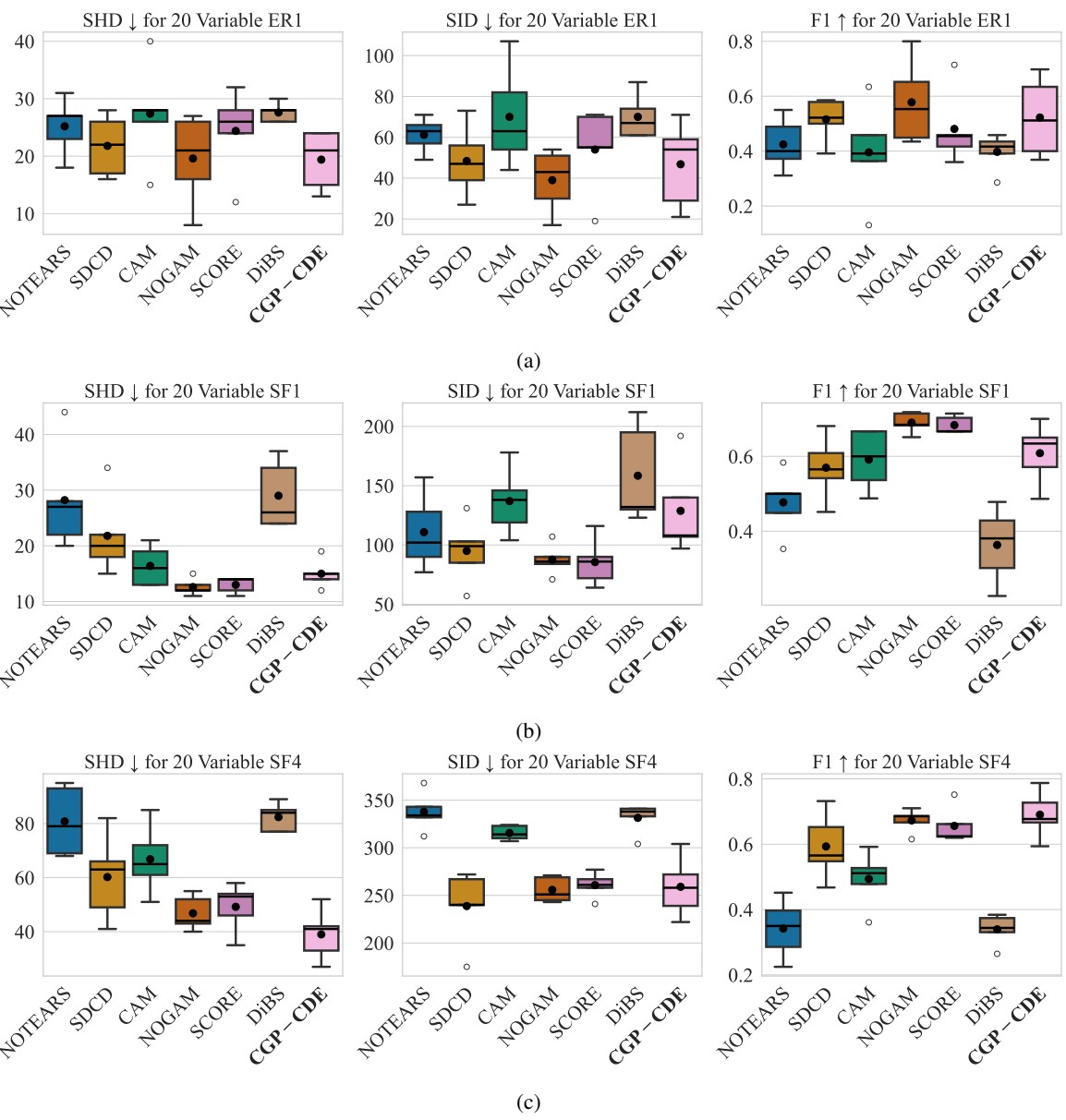

Figure 6: (a) Erdos-Renyi (ER) graphs with one expected edge per variable. (b) Scale-Free (SF) graphs with one expected edge per variable. (d) SF graphs with four expected edges per variable. For results for Erdos-Renyi (ER) graphs with four expected edges per variable, see Figure 4b. Black dot is the average metric. Lower is better for SHD and SID, higher is better for F1.

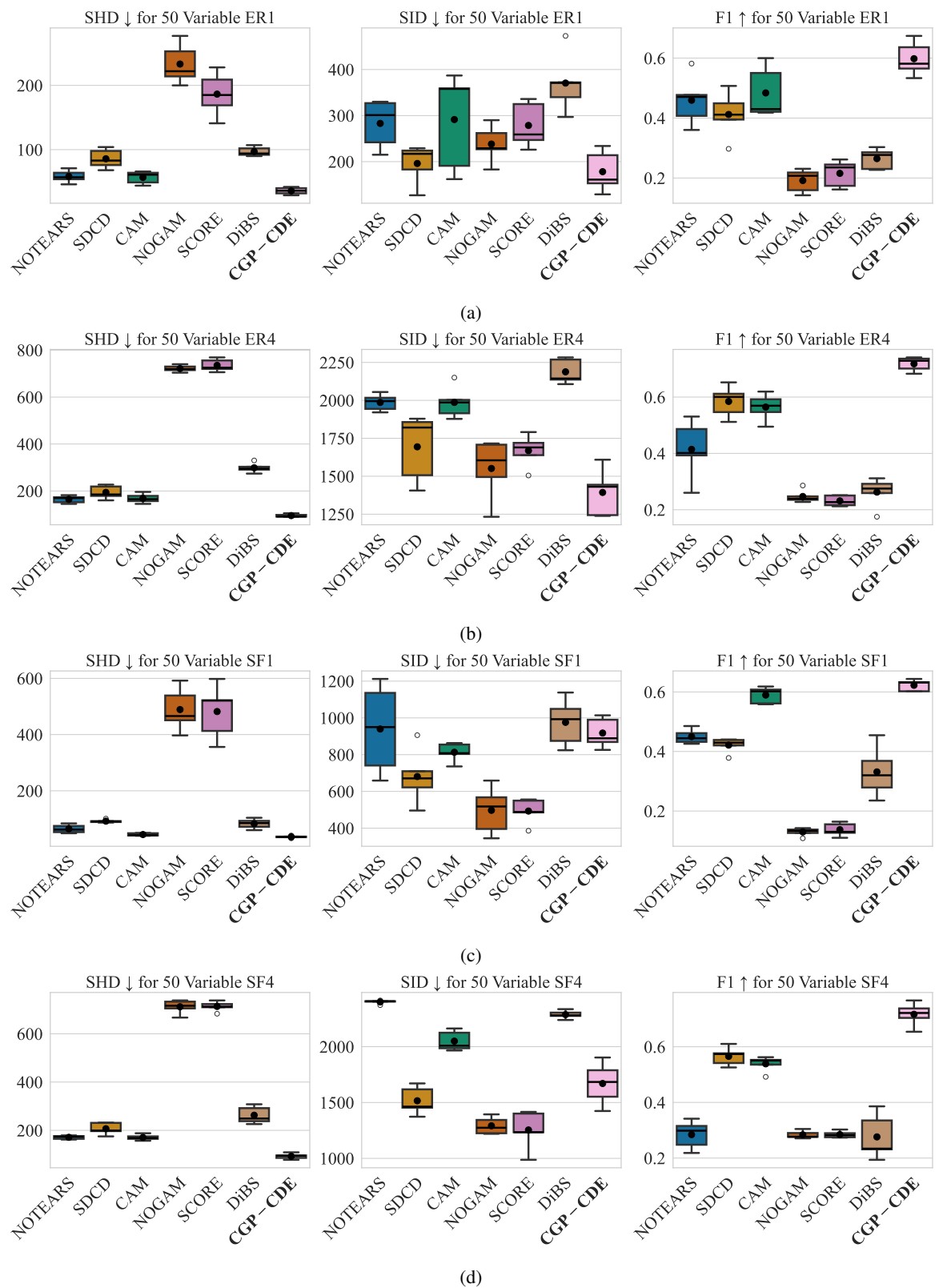

Figure 7: (a) Erdos-Renyi (ER) graphs with one expected edge per variable. (b) ER graphs with four expected edges per variable. (c) Scale-Free (SF) graphs with one expected edge per variable. (d) SF graphs with four expected edges per variable. Black dot is the average metric. Lower is better for SHD and SID, higher is better for F1.

