# OpenReview forum: "Continuous Bayesian Model Selection for Multivariate Causal Discovery"
_ICML.cc/2025/Conference — ICML 2025 poster_

### Official Review · Reviewer_7Mfs · 2025-03-08

**Overall Recommendation:** 3

**Summary:**

This paper studies structure learning for observational data using Bayeisan model selection. It falls into the category of score based learning and uses model evidence as score to select DAG. It shows the existing work on bivariate case [Dhir et al., 2024] can be extended to multivariate case, and applies a flexible model called Causal Gaussian Process Conditional Density Estimator (CGP-CDE) with continuous reparametrization and variational inference to learn the DAG. Experiments are conducted to show the effectiveness.

**Claims And Evidence:**

The claims on improvement are supported via experiments.

**Essential References Not Discussed:**

No

**Experimental Designs Or Analyses:**

The experimental design makes sense.

**Methods And Evaluation Criteria:**

The evaluation criteria via SHD, SID, F1 are a commonly used critera.

**Other Comments Or Suggestions:**

- Many mathemetical details are hidden in the appendix. Especially for Section 3 on the theoretical extension, there is no one theoretical statement in this section while everything is burried in the appendix.

**Other Strengths And Weaknesses:**

- The paper derives from the ICM assumption for identifiability and allows for much flexible class of identifiable Bayesian network
- The proposed method leverages the powerful tools for modelling and learning nonparametric DAGs in a scalable manner.
- The experiments show competitive performance with many benchmarks.

**Questions For Authors:**

- The definition of Bayesian equivalence (identifiability and distinguishabllity) involves data $X$, also relates to the definition of probrobability of error. I wonder for two models that are not Bayesian equivalent, why can they have positive probrobability of error. Should it be consistent to use the model evidence to find the true DAG? I guess the question is does the sample size in the definition of Bayesian equivalence goes to infinity? Or why do we have identifiability and distinguishabllity and what are their differences?

**Relation To Broader Scientific Literature:**

Causal discovery is widely used in scientific research for exploratory analysis. Developing identifiability and scalable method is important for broader scientific literature.

**Theoretical Claims:**

One of the proofs on the main results --  Theorem B.6 -- concerns me. One of the main contribution of this paper is to extend the existing identifiability result to multivariate case. However, the proof of this theorem only says it follows directly from [Dhir et al., 2024] without any details.

---

> ### Author Rebuttal · Authors · 2025-03-31
>
> Thank you for your positive and encouraging feedback on our work. We appreciate your acknowledgement that the proposed method **"allows for learning nonparametric DAGs in a scalable manner"** and that our **"experiments show competitive performance with the benchmarks"**. We address your comments in the following.
>
> >However, the proof of this theorem only says it follows directly from [Dhir et al., 2024] without any details.
>
> Our construction and proofs earlier in the sections lead us to write the first part of our theorem exactly as Prop. 4.7 in Dhir et al., 2024. Hence, we can directly refer to the proof. We will make this clearer by rewriting the theorem from Dhir et al., (2024) and explicitly stating where we invoke the result.
>
> > Many mathemetical details are hidden in the appendix. Especially for Section 3 on the theoretical extension, there is no one theoretical statement in this section while everything is burried in the appendix.
>
> We would love to put more details in the main paper. Our theory requires us to define several concepts not relevant to the rest of the paper, but that allow us to state accurate and precise theorems. As this takes up a lot of space, we thought it best to do this in the Appendix. **We do provide a general intuition for our theoretical results in the main paper** (L174 LHS). We hope you agree with this approach.
>
> > I wonder for two models that are not Bayesian equivalent, why can they have positive probrobability of error
>
> The posteriors may overlap even in the infinite data setting. We can show this with a very simple example. Normalised linear Gaussian models are not identifiable (in the population setting) [1, Appendix D.2]. If a chosen model can approximate a normalised linear Gaussian model, and we sample from the model, there is a non-zero probability that we sample a normalised linear Gaussian dataset. Thus the **chosen model must have non-zero probability of error**.
> Hence, while non-linear additive noise models are identifiable in general, additive noise models (also containing linear functions) will have a positive probability of error (but not completely unidentifiable).
>
> [1] Dhir et al., "Bivariate Causal Discovery using Bayesian Model Selection." ICML, 2024.
>
> [2] Hoyer et al. "Nonlinear causal discovery with additive noise models." Advances in neural information processing systems 21 (2008).
>
> > Should it be consistent to use the model evidence to find the true DAG? I guess the question is does the sample size in the definition of Bayesian equivalence goes to infinity?
>
> The definition of Bayesian distribution-equivalence holds for any sample size. However, our theorems (B.3 and B.6) hold in the population setting. We do state that in B.3, but we will state that in theorem B.6. Thank you for pointing this out.
>
> >Or why do we have identifiability and distinguishabllity and what are their differences?
>
> Identifiability is where the probability of error is zero, whereas distinguishability is where the probability of error is less than random uniform. We will make it clear that these are in the population setting where we define these concepts in L200 LHS.

---

> > ### Comment · Reviewer_7Mfs · 2025-04-04
> >
> > I thanks the authors for their response, which has assured some of my concerns. I keep the score unchanged to reflect my lack of familiarity.

---

### Official Review · Reviewer_2axY · 2025-03-10

**Overall Recommendation:** 2

**Summary:**

This paper presents a multivariate causal discovery approach based on Bayesian model selection. It builds on the work of Dhir et al. (2024), who proposed to use Bayesian model selection to identify causal direction in the bivariate case. The Bayesian model selection framework allows for a trade-off between a model's goodness of fit and complexity. The original work showed that Bayesian Model Selection can discriminate causal directions even when Maximum Likelihood (ML) methods fail due to distribution equivalence, thanks to the marginal likelihood, which, unlike the likelihood, is not symmetric (they consider the whole function space instead of the best-fitting function).
They use the Causal Gaussian Process Conditional Density Estimator (CGP-CDE) to impose no restrictions on the model (e.g. linear model, additive noise...), which is a significant contribution of this work. This makes it possible to relax the assumptions, one of the most critical challenges in causal discovery. On the other hand, their method does not come with a strict identifiability guarantee due to potential overlaps in the posterior of different causal models. They propose a way to estimate the error probability using a sampling strategy.
The causal identifiability proof is based on the Independent Causal Mechanism (ICM) assumption, which the authors use to claim that well-chosen Bayesian priors for the distribution of a cause and the distribution of an effect given the cause should be independent.
In practice, to avoid comparing all possible DAGs and their posteriors, the model is identified by continuously optimizing the hyperparameters of the Gaussian process priors using a variational autoencoder (VAE) with a Bayesian model selection-based loss and acyclicity penalty. They also define priors on the resulting directed acyclic graphs (DAGs) to favor sparse models. As with most existing methods, their approach only works if all confounders are observed (causal sufficiency assumption).

**Claims And Evidence:**

Not all claims are theoretically substantiated. The authors claim to find a causal model. Their continuous optimization method yields a direct acyclic graph, but it is not made clear whether the found model respects the Markov condition (each node is conditionally independent of its non-descendants, given its parents), which is required for a DAG to be causal. It is unclear whether the continuous optimization method finds well-chosen priors, which is required for identifiability.

**Essential References Not Discussed:**

The authors do not discuss the literature on the information-theoretic view of causality (Janzing, Schölkopf, 2010) in the related works although it is conceptually very close -- instantiating Occam's razor to distinguish between models. There is no comparison with other score-based approaches that also aim to balance complexity and goodness of fit, such as GES (Chickering, Maxwell, 2002), which uses the Bayesian Information Criterion (BIC), and GLOBE (Mian, 2021), which uses a two-part Minimum Description Length (MDL) score.

References:
- Chickering, David Maxwell. "Optimal structure identification with greedy search." Journal of machine learning research 3.Nov (2002): 507-554.
- Mian, Osman A., Alexander Marx, and Jilles Vreeken. "Discovering fully oriented causal networks." Proceedings of the AAAI Conference on Artificial Intelligence. Vol. 35. No. 10. 2021.
- Janzing, Dominik, and Bernhard Schölkopf. "Causal inference using the algorithmic Markov condition." IEEE Transactions on Information Theory 56.10 (2010): 5168-5194.

**Experimental Designs Or Analyses:**

The experimental design includes several settings. Results on synthetic data generated with a neural network show that the proposed method largely outperforms competitors, which was expected since their assumptions (e.g. additive noise) are violated. However, the author(s) also provide results for data generated with additive noise and their performance is still competitive with other methods, showing that the relaxation of the assumptions does not come at the expense of overall performance.

**Methods And Evaluation Criteria:**

The synthetic data experiment is performed with different generation settings (neural network as function, noise additive or not...). Syntren is a simulated data set. There is no experiment on real data. SHD and SID are commonly used metrics for causal discovery.

**Other Comments Or Suggestions:**

NOTEARS is a structure learning approach, not a causal discovery algorithm. That is, it finds a DAG, but there are no guarantees this a causal one. This should be mentioned. I have the same concern about the proposed method.


### update after rebuttal ###

I thank the authors for their answers, but I'm afraid they did not take away my concerns regarding the identifiability of the approach and therewith whether this is a method that is guaranteed to return causal structures.

**Other Strengths And Weaknesses:**

Strengths:
- The paper is well written and well motivated.
- The use of a flexible model and VAE allows for fewer assumptions than usual and therefore makes the model applicable to a wider range of applications.
- The theory is well covered and convincing.
- The extensive experiments show that the proposed methods significantly outperform the evaluated competitors.

Weaknesses:
- The causal model is not strictly identifiable.
- Continuous optimisation may not be efficient (see questions 1, 2 and 3).
- It is unclear whether the continuous optimisation approach maintains the theoretical guarantees (see questions 4 and 5).

**Questions For Authors:**

1) How do you interpret the poor performance of CGP-CDE in the 3-variable experiment?

2) In the continuous optimization, the warm-up phase seems computationally expensive (about 25,000 iterations for 1,000 samples?). Is this a typical number of iterations for an initialization phase? Could another initialization reduce the number of iterations needed? What about the number of iterations for the cooling phase (also 25,000 iterations for 1,000 samples)?

3) In the 3-variable experiment, where the data are generated using a Bayesian causal model, DGP-CDE acts as a sanity check, as it considers all models and thus includes the true one used to generate the data.  However, the poor performance of the continuous optimization is worrying given the small search space (25 graphs) and the fact that the data was generated using a Bayesian causal model (unlike the opponent whose assumptions were violated). How do you explain this?

**Relation To Broader Scientific Literature:**

This paper contributes to the field of causal discovery and, more specifically, to the score-based and continuous optimisation-based lines of research. It extends the work of Dhir et al. (2024), who proposed using Bayesian model selection for causal edge orientation. This paper's novelty is its adaptation and proof for the multivariate case, the use of CGP-CDE for flexible modeling, and the continuous optimization of model parameters to find a DAG.

**Theoretical Claims:**

I read the proofs in the appendix and found no errors.

---

> ### Author Rebuttal · Authors · 2025-03-31
>
> Thank you for your insightful review. We appreciate your recognition of our contribution to the field of causal discovery. We are glad you think the **"theory was well covered and convincing"**, the paper is **"well written and well motivated"** and our **"extensive experiments significantly outperform the evaluated competitors"**.
>
> > whether the found model respects the Markov condition
>
> **Our model itself satisfies the causal Markov condition at the end of training by construction**. This happens in the acyclicity thresholding phase of the training (L935). The causal Markov assumption follows trivially from the assumption of no hidden confounders (L82 LHS) and that the noise terms are independent (eq 8).
>
> We are aware of the larger discussion of whether the causal Markov condition implies conditional independences in all generality [1]. However, the same work states that the added assumption of "Modularity" (Property 7.3 in [1]) provides a "fully quantitative solution to the problem of inferring causality from observational data". The ICM assumption we make (L145 LHS) implies a form of modularity [2]. We note that works like NOTEARS do not make this assumption, and the only similarity to our work is that we also use a continuous optimisation scheme.
>
> [1] Dawid et al., "Beware of the DAG!." Causality: objectives and assessment. PMLR, 2010.
>
> [2] Janzing et al., "Causal inference using the algorithmic Markov condition." IEEE Transactions on Information Theory (2010).
>
> > It is unclear whether the continuous optimization method finds well-chosen priors
>
> Our continuous optimization does not find the prior, but the *posterior* given a prior (L255 RHS).
>
> > literature on the information-theoretic view of causality (Janzing, Schölkopf, 2010)
>
> The basic principle that allows for distinguishing causal structure is very related to the work you have mentioned [1, Appendix B.2]. We will include a discussion on this.
>
> [1] Dhir et al., "Bivariate Causal Discovery using Bayesian Model Selection." ICML, 2024.
>
> > There is no comparison with other score-based approaches
>
> We have run GES and GLOBE for all experiments, results can be viewed here: https://anonymous.4open.science/r/Additional-Results/. GES performs poorly compared to CGP-CDE and the other baselines, except for competitve performance on the 50 var ER1 dataset. This is because GES performs well on sparse graphs, but struggles with denser graphs as it gets stuck in local optima. GLOBE didn't perform well on most of the datasets, except getting the best SHD (but poor SID and F1) for Syntren.
>
> > The causal model is not strictly identifiable.
>
> This is true, but we show that **for our model the probability of error is small**. Our motivation is that restrictions made to gain strict identifiability can be unrealistic. When the data is generated from a different model, identifiability does not hold anyway. We argue that tolerating a small probability of error for gaining more flexibility can be beneficial in these cases. We show exactly this in our experiments.
>
> > How do you interpret the poor performance of CGP-CDE in the 3-variable experiment?
>
> We discuss this in Line 368 RHS, although we note the wrong figure was referenced, which we will fix. The good performance of the discrete case **shows that our objective empirically identifies the correct causal graph in the multivariate case**. The continuous approximation uses (noisy) gradients to find the most likely causal structure, which introduces errors.
> The comparison to the continuous case thus quantifies the error introduced by the continuous approximation. We believe this shows that the continuous approximation scheme can be improved on. Nevertheless, we emphasise that we clearly outperform baselines in datasets with more variables (Appendix I).
>
> > Could another initialization reduce the number of iterations needed?
>
> We were overly conservative with the number of iterations for the warm up and cool down phases. You are correct, better initialisation will reduce the number of iterations required. This can be monitored simply by looking at the training curve.
>
> > In the 3-variable experiment... How do you explain this?
>
> The point of this experiment was to **show the price we pay for scalability**. The difference in performance is due to the continuous optimisation, which introduces errors (see above). These results show that while our Bayesian principle is correct, large improvements can be made in the continuous approximation scheme. As we experiment on **all possible** 3 variable graphs, this experiment also contains a more dense graphs (relative to the number of variables), which may be harder to find.
>
> Note we also provide other experiments with more variables where the data isn't generated from our model and the ANM experiments where the baseline models' assumptions aren't violated. In both cases, we outperform the baselines.
>
> The reviewer refers to questions 4 and 5 but the questions only go up to 3.

---

### Official Review · Reviewer_Ry2n · 2025-03-14

**Overall Recommendation:** 3

**Summary:**

Recent work shows that in the bivariate case Bayesian model selection can be used for structure identification under more flexible assumptions at the cost of a small probability of error. This paper extends the previous result to the multivariate case. The authors empirically validate the method by comparing to multiple baselines under extensive experimental settings.

## update after rebuttal

Thank you for the author's response. After reading all the review comments I decide to keep my rating unchanged.

**Claims And Evidence:**

Yes.

**Essential References Not Discussed:**

Related works are properly discussed.

**Experimental Designs Or Analyses:**

Yes. The experimental designs look sensible to me. The setting is extensive including both synthetic setting with multiple variables and real-life experiments.

**Methods And Evaluation Criteria:**

It makes sense to me that for two distribution equivalent graphs we can rely on independent priors to distinguish them. Yet, it is unclear to me that do we need to know the correct prior in advance? If yes, what will happen under misspecification of prior?

**Other Comments Or Suggestions:**

N.A.

**Other Strengths And Weaknesses:**

Strengths

1. The proposed method and theory of using bayesian model selection for multivariate structure learning look novel to me.

2. The experimental setting is extensive including both synthetic and real-life data.

3. It is interesting to see methods that can distinguish between graphs that are distribution-equivalent.

Weaknesses

1.  I am not very familiar with bayesian model selection so I am not sure about the significance of extending from bivariate (Dhiretal.,2024) to the multivariate scenario. I will defer to other reviewers regarding this point.

**Questions For Authors:**

Please refer to my question in Methods And Evaluation Criteria part.

**Relation To Broader Scientific Literature:**

Key contributions of the paper compared to prior work are properly discussed.

**Theoretical Claims:**

No.

---

> ### Author Rebuttal · Authors · 2025-03-31
>
> Thank you for your detailed feedback. We appreciate your acknowledgement of the **novelty of our approach in applying Bayesian model selection for multivariate structure learning** and your recognition of the **thorough discussion of our contributions relative to prior work**. We are also pleased you found the **experimental design sensible and extensive**. Additionally, we are glad you highlighted the importance of our method’s ability to distinguish between distribution-equivalent graphs, which is a key strength of our approach. We address your specific comments in the following.
>
> > do we need to know the correct prior in advance?
>
> It is not necessary to know the correct prior in advance. **A key reason that allows for distinguishing causal graphs is the ICM assumption that is encoded in the prior** (L153 LHS, "separable compatible" in Theorem B.6). The prior over functional mechanisms is also important. Our approach for this was to try and choose a model/prior that ensures as much as possible we do not put zero probability on any dataset (see for example [1, Section 1.2]).
>
> [1] Hjort et al., eds. Bayesian nonparametrics. Vol. 28. Cambridge University Press, 2010.
>
> > what will happen under misspecification of prior?
>
> We discuss this in L183 RHS. The prior defines what datasets are likely under our model (data distribution). This data distribution can be used to define a probability of error, which is a measure of how distinguishable causal graphs under our model are (eq 7). Given datasets from a *separate* distribution (data generating process), the accuracy of the estimated probability of error depends on how far the model's data distribution is from the data generating process. Small differences in the model and the data generating process do not result in complete invalidity of the estimated probability of error. As with any method, large variations will mean the estimated probability of error differs from the true probability of error (of the data generating process). For more discussion on this see [1, Section 4.4].
>
> We note that the a-priori functional restrictions imposed by previous methods are also unverifiable assumptions. As with any unverifiable assumption, it is necessary to empirically validate how well it works in practice. We test exactly this in our experiments (Section 6.2, 6.3), **where the data generating processes don't match our model prior and our model outperforms the baseline methods**.
>
> [1] Dhir et al., "Bivariate Causal Discovery using Bayesian Model Selection." ICML, 2024.
>
> > I am not very familiar with bayesian model selection so I am not sure about the significance of extending from bivariate (Dhiretal.,2024) to the multivariate scenario. I will defer to other reviewers regarding this point.
>
> Dhir et al.,2024 consider the bivariate case, but we answer the question: does the theory hold in the multivariate case, and how does the performance scale with number of variables?
> We theoretically show Bayesian model selection works for multivariate datasets and propose a method to effectively scale to large numbers of variables, overcoming the costs which would be super-exponential with number of variables.
> We then show that this approach outperforms competing methods in the multivariate case.

---

### Official Review · Reviewer_apg4 · 2025-03-14

**Overall Recommendation:** 4

**Summary:**

The paper proposes a new method called CGP-CDE for (Bayesian) causal model discovery that allows for less restrictive model assumptions and can be applied to higher dimensional systems as well. It is based on a GP approach to obtain a nonparametric conditional density estimator for each node given its parents in the causal DAG, ultimately capturing the likelihood of a graph. The search for the best fitting model is turned into a continuous optimization problem by adding a familiar acyclicity constraint with penalty on the weight matrix representing the graph. The method is evaluated on synthetic/realistic data and found to compare equal or favourably to other alternatives.

## update after rebuttal
I thank the authors for their reply, and, having read the other reviews & rebuttals as well, I am happy to leave my score at '4: accept'.

One final remark: I understand the temptation to go for maximal informativeness, but the authors will know that typically the score differences within a MEC are much smaller than between MECs, and therefore the extra information from orienting the full graph tends to be much less reliable, in turn making the entire output less trustworthy = less useful in practice. Given that any directed graph has a unique mapping to a MEC one does not need to 'remove' the preference within a MEC. Instead, we can easily show both outputs, but with a much higher reliability for anything implied by the MEC representation, giving practitioners an intuitive way to distinguish between more and less reliable conclusions in the output.

**Claims And Evidence:**

The paper initially suggests that it will solve the problem of restrictive / unrealistic model assumptions encountered when tackling real world data, but this is of course nonsense. It still starts from the causal sufficiency and acyclicity assumptions (which is not how the world works), and relies on a rather arbitrary prior to go from Markov equivalence class (MEC) to unique model. Yes it ensures identifiability, but you are essentially finding back what you put in and assume/hope that it matches reality.
However, I do really like the GP approach to modelling the likelihood, which indeed does provide a significant improvement over existing parametric approaches to Bayesian/likelihood based inference.

**Essential References Not Discussed:**

Cooper & Herskovitz (1992): this old but seminal work already contains a Bayesian score-based method that relies on the prior to select an optimal posterior causal DAG. The subsequent work by Heckerman showed how to combat this (rather undesirable) behaviour by introducing a score that ensures MEC-equivalence, but the current paper now seems to suggest that they came up with the idea of selecting models *within* a MEC by using a non-equivalent score …

**Experimental Designs Or Analyses:**

As mentioned, experimental evaluation is limited but ok. One of the most striking findings is that the discrete version (DGP-CDE) is vastly superior (FIg.1), but that the authors still choose to focus on the (currently in vogue) continuous approximation, even though it suffers from the exact same problems w.r.t. finding the global optimum in anything with higher dimensions. But I will not hold that against the authors :)

**Methods And Evaluation Criteria:**

Experimental evaluation is limited but ok, and demonstrates the potential of the core contribution.

**Other Comments Or Suggestions:**

No

**Other Strengths And Weaknesses:**

As stated: paper suggests more than it does, and some aspects (like aiming for unique identifiability from using an in practice hard to justify/assess prior) actually weakens the output, and I would have preferred§ a modest but more robust aim for e.g. the MEC.
Also starting from the causal sufficiency assumption (and to some degree also the acyclicity assumption) is `unforgivable’ when the aim is to make a causal discovery method more suitable to real-world applications. Yes it makes everything easier, but that is not a good enough reason to do / keep doing it.

On the plus side: paper is clear and well written, and the GP approach at the heart of the method is a significant and promising contribution, and for that reason I will recommend accept.

**Questions For Authors:**

No

**Relation To Broader Scientific Literature:**

Relevant (recent) work is discussed.

**Theoretical Claims:**

No explicit theoretical claims in the main paper, but overall approach is sound.

---

> ### Author Rebuttal · Authors · 2025-03-31
>
> Thank you for your positive and encouraging feedback. We are glad you found the paper **"clear and well written"**, and appreciate your comment that our method is a **"significant and promising contribution"**.
>
> > The paper initially suggests that it will solve the problem of restrictive / unrealistic model assumptions encountered when tackling real world data
>
> Previous methods regularly make restrictive functional assumptions that may not hold in practice. In this paper, **we address relaxing these restrictive functional modelling assumptions** (L12 RHS, L172 LHS).
> We wholeheartedly agree causal sufficiency and acyclicity can be unrealistic assumptions and are worth relaxing. These are very common in causal discovery algorithms, and the baselines we compare against also make these assumptions. We hope we didn't overclaim on this point and will make clear that this paper is only a step towards more realistic causal discovery (L70 LHS).
>
> > Yes it ensures identifiability, but you are essentially finding back what you put in and assume/hope that it matches reality.
> > ...using an in practice hard to justify/assess prior
>
> Our work shows that a relatively simple assumption - independent causal mechanisms (see L148 LHS) - can allow for distinguishing causal structure within a Markov equivalence class (L129 RHS, Theorem B.6). This assumption can be encoded in the prior by ensuring the priors factorise appropriately (L153 LHS).
>
> The specific priors over functional mechanisms are also important.
> Our approach for this was to ensure, as much as possible, that the prior does not put zero probability on any dataset (see for example [1, Section 1.2]). The model/prior that we chose is not arbitrary, but close to known identifiable models (non-linear additive noise models), as shown by the good performance on additive noise datasets (Appendix I.1). The difference to previous methods is that we do not make hard restrictions - our model allows us to approximate more than just additive noise datasets (L232 LHS).
>
> We note that assumptions made in previous methods that a-priori restrict functional form are also unverifiable. Any unverifiable assumption requires empirical verification. This is exactly what we do in sections 6.2 and 6.3. Here, the data is generated from mechanisms that are different to all the models. We vary other factors as well (graph types and density). Our method outperforms previous methods (full results in Appendix I). We believe this shows the usefulness of the Bayesian approach.
>
> [1] Hjort et al., eds. Bayesian nonparametrics. Vol. 28. Cambridge University Press, 2010.
>
> > One of the most striking findings is that the discrete version (DGP-CDE) is vastly superior (FIg.1), but that the authors still choose to focus on the (currently in vogue) continuous approximation, even though it suffers from the exact same problems w.r.t. finding the global optimum in anything with higher dimensions.
>
> This is indeed very interesting. The cost of the discrete version, which requires enumerating over all possible graphs, is too high for more than a few variables. We included this result because the difference in performance between the CGP-CDE and the DGP-CDE clearly shows that, although our principle is correct, there is room for improvement in the continuous relaxation. However, CGP-CDE allows us to scale to a larger numbers of variables.
>
> > Cooper & Herskovitz (1992): this old but seminal work already contains a Bayesian score-based method that relies on the prior to select an optimal posterior causal DAG...
>
> The papers you mention are important but only consider simple linear models.
> However, it has been known that with more complicated models, Bayesian models tend to have an opinion within an MEC [1,2,3]. The main reason (as shown in Appendix B) is because to create equivalent models, the ICM assumption has to hold in multiple factorisations (L129 RHS, Theorem B.6). While it is relatively simple to construct models that do this with linear models [1, Appendix D.1, D.2], it is not clear whether this is possible with more complex models [1, Appendix D.3].
>
> [1] Dhir et al., "Bivariate Causal Discovery using Bayesian Model Selection." ICML, 2024.
>
> [2] Friedman et al., "Gaussian process networks." Proceedings of the Sixteenth conference on Uncertainty in artificial intelligence. 2000.
>
> [3] Stegle et al. "Probabilistic latent variable models for distinguishing between cause and effect." Advances in neural information processing systems 23 (2010).
>
> > I would have preferred a modest but more robust aim for e.g. the MEC.
>
> This would obscure the fact that the model (due to the ICM assumption in the prior) **has a preference** for certain causal structures within an MEC over others (Line 138 RHS and line 720). **This preference within an MEC can only be removed by breaking the ICM assumption** (Theorem B.6). We thus make use of this preference.

---

### Decision · Program_Chairs · 2025-05-01

**Decision:**

Accept (poster)

**Comment:**

This paper primarily extends Dhir et al. (2024) from the bivariate to the multivariate setting, using the old idea (as one reviewer points out, this dates back at least to Cooper & Herskovitz, 1992) of letting priors select causal between models that would otherwise be equivalent. The key contribution is to propose a new differentiable NOTEARS-style model under a nonlinear Gaussian process model for the conditional densities (CGP-CDE) that is more flexible. The authors show that their method outperforms recent methods based on differentiable causal discovery.

An important part of this line of work is to make the methods and evaluation open-source. During the discussion, the authors clarified that they "plan to make the Github repo containing the code public upon publication", which I expect the authors will do, with prominent links to the repo in the camera ready.